# RotoGrad: Gradient Homogenization in Multitask Learning

**Adrián Javaloy**
Department of Computer Science
Saarland University
Saarbrücken, Germany
ajavaloy@cs.uni-saarland.de

**Isabel Valera**
Department of Computer Science
Saarland University
Saarbrücken, Germany

## Abstract

Multitask learning is being increasingly adopted in applications domains like computer vision and reinforcement learning. However, optimally exploiting its advantages remains a major challenge due to the effect of negative transfer. Previous works have tracked down this issue to the disparities in gradient magnitudes and directions across tasks when optimizing the shared network parameters. While recent work has acknowledged that negative transfer is a two-fold problem, existing approaches fall short. These methods only focus on either homogenizing the gradient magnitude across tasks; or greedily change the gradient directions, overlooking future conflicts. In this work, we introduce RotoGrad, an algorithm that tackles negative transfer as a whole: it jointly homogenizes gradient magnitudes and directions, while ensuring training convergence. We show that RotoGrad outperforms competing methods in complex problems, including multi-label classification in CelebA and computer vision tasks in the NYUv2 dataset. A Pytorch implementation can be found in https://github.com/adrianjav/rotograd.

## 1 Introduction

As neural network architectures get larger in order to solve increasingly more complex tasks, the idea of jointly learning multiple tasks (for example, depth estimation and semantic segmentation in computer vision) with a single network is becoming more and more appealing. This is precisely the idea of multitask learning (MTL) (Caruana, 1993), which promises higher performance in the individual tasks and better generalization to unseen data, while drastically reducing the number of parameters (Ruder, 2017).

Unfortunately, sharing parameters between tasks may also lead to difficulties during training as tasks compete for shared resources, often resulting in poorer results than solving individual tasks, a phenomenon known as *negative transfer* (Ruder, 2017). Previous works have tracked down this issue to the two types of differences between task gradients. First, *differences in magnitude* across tasks can make some tasks dominate the others during the learning process. Several methods have been proposed to homogenize gradient magnitudes such as MGDA-UB (Sener & Koltun, 2018), GradNorm (Chen et al., 2018), or IMTL-G Liu et al. (2021b). However, little attention has been put towards the second source of the problem: *conflicting directions* of the gradients for different tasks. Due to the way gradients are added up, gradients of different tasks may cancel each other out if they point to opposite directions of the parameter space, thus leading to a poor update direction for a subset or even all tasks. Only very recently a handful of works have started to propose methods to mitigate the conflicting gradients problem, for example, by removing conflicting parts of the gradients (Yu et al., 2020), or randomly 'dropping' some elements of the gradient vector (Chen et al., 2020).

In this work we propose RotoGrad, an algorithm that tackles negative transfer as a whole by homogenizing both gradient magnitudes and directions across tasks. RotoGrad addresses the gradient magnitude discrepancies by re-weighting task gradients at each step of the learning, while encouraging learning those tasks that have converged the least thus far. In that way, it makes sure that no task is overlooked during training. Additionally, instead of directly modifying gradient directions, RotoGrad smoothly rotates the shared feature space differently for each task, seamlessly aligning gradients in

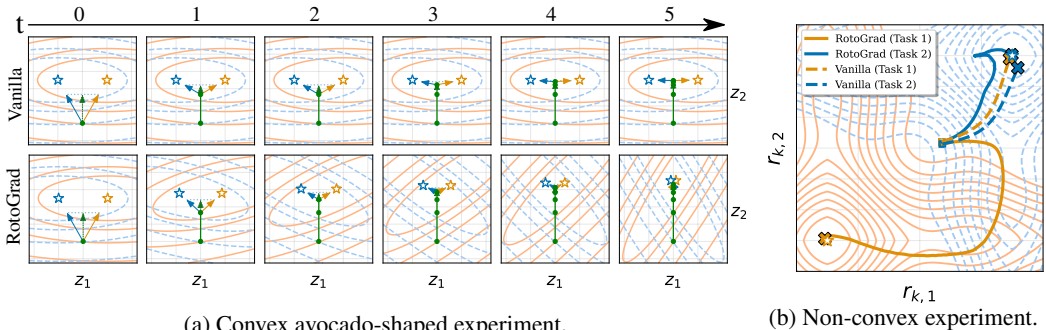

(a) Convex avocado-shaped experiment.  (b) Non-convex experiment.

Figure 1: Level plots showing the evolution of two regression MTL problems with/without RotoGrad, see §4. RotoGrad is able to reach the optimum (☆) for both tasks. *(a)* In the space of $z$, RotoGrad rotates the function-spaces to align task gradients (blue/orange arrows), finding shared features $z$ (green arrow) closer to the (matched) optima. *(b)* In the space of $r_k$, RotoGrad rotates the shared feature $z$, providing per-task features $r_k$ that better fit each task.

the long run. As shown by our theoretical insights, the cooperation between gradient magnitude- and direction-homogenization ensures the stability of the overall learning process. Finally, we run extensive experiments to empirically demonstrate that RotoGrad leads to stable (convergent) learning, scales up to complex network architectures, and outperforms competing methods in multi-label classification settings in CIFAR10 and CelebA, as well as in computer vision tasks using the NYUv2 dataset. Moreover, we provide a simple-to-use library to include RotoGrad in any Pytorch pipeline.

## 2 MULTITASK LEARNING AND NEGATIVE TRANSFER

The goal of MTL is to simultaneously learn $K$ different tasks, that is, finding $K$ mappings from a common input dataset $\boldsymbol{X} \in \mathbb{R}^{N \times D}$ to a task-specific set of labels $\boldsymbol{Y}_k \in \mathbb{Y}_k^N$. Most settings consider a hard-parameter sharing architecture, which is characterized by two components: the *backbone* and *heads* networks. The backbone uses a set of shared parameters, $\boldsymbol{\theta}$, to transform each input $\boldsymbol{x} \in \boldsymbol{X}$ into a shared intermediate representation $\boldsymbol{z} = f(\boldsymbol{x}; \boldsymbol{\theta}) \in \mathbb{R}^d$, where $d$ is the dimensionality of $\boldsymbol{z}$. Additionally, each task $k = 1, 2, \ldots, K$ has a head network $h_k$, with exclusive parameters $\boldsymbol{\phi}_k$, that takes this intermediate feature $\boldsymbol{z}$ and outputs the prediction $h_k(\boldsymbol{x}) = h_k(\boldsymbol{z}; \boldsymbol{\phi}_k)$ for the corresponding task. This architecture is illustrated in Figure 2, where we have added task-specific rotation matrices $\boldsymbol{R}_k$ that will be necessary for the proposed approach, RotoGrad. Note that the general architecture described above is equivalent to the one in Figure 2 when all rotations $\boldsymbol{R}_k$ correspond to identity matrices, such that $\boldsymbol{r}_k = \boldsymbol{z}$ for all $k$.

MTL aims to learn the architecture parameters $\boldsymbol{\theta}, \boldsymbol{\phi}_1, \boldsymbol{\phi}_2, \ldots, \boldsymbol{\phi}_K$ by simultaneously minimizing all task losses, that is, $L_k(h_k(\boldsymbol{x}), \boldsymbol{y}_k)$ for $k = 1, \ldots, K$. Although this is a priori a multi-objective optimization problem (Sener & Koltun, 2018), in practice a single surrogate loss consisting of a linear combination of the task losses, $L = \sum_k \omega_k L_k$, is optimized. While this approach leads to a simpler optimization problem,

$$
\boldsymbol{x} \xrightarrow{f_{\boldsymbol{\theta}}} \boldsymbol{z}
\begin{cases}
\boldsymbol{R}_1 \to \boldsymbol{r}_1 \xrightarrow{h_{\phi_1}} L_1(h_1(\boldsymbol{r}_1), \boldsymbol{y}_1) \\
\boldsymbol{R}_2 \to \boldsymbol{r}_2 \xrightarrow{h_{\phi_2}} L_2(h_2(\boldsymbol{r}_2), \boldsymbol{y}_2) \\
\quad\cdots \qquad\qquad\qquad \cdots \\
\boldsymbol{R}_K \to \boldsymbol{r}_K \xrightarrow{h_{\phi_K}} L_K(h_K(\boldsymbol{r}_K), \boldsymbol{y}_K)
\end{cases}
$$

Figure 2: Hard-parameter sharing architecture, including the rotation matrices $\boldsymbol{R}_k$ of RotoGrad.

it may also trigger *negative transfer* between tasks, hurting the overall MTL performance due to an imbalanced competition among tasks for the shared parameters (Ruder, 2017).

The negative transfer problem can be studied through the updates of the shared parameters $\boldsymbol{\theta}$. At each training step, $\boldsymbol{\theta}$ is updated according to a linear combination of task gradients, $\nabla_{\boldsymbol{\theta}} L = \sum_k \omega_k \nabla_{\boldsymbol{\theta}} L_k$, which may suffer from two problems. First, **magnitude differences** of the gradients across tasks may lead to a subset of tasks dominating the total gradient, and therefore to the model prioritizing them over the others. Second, **conflicting directions** of the gradients across tasks may lead to update directions that do not improve any of the tasks. Figure 1 shows an example of poor direction updates (left) as well as magnitude dominance (right).

In this work, we tackle negative transfer as a whole by homogenizing tasks gradients both in magnitude and direction. To reduce overhead, we adopt the usual practice and homogenize gradients with respect to the shared feature $z$ (rather than $\theta$), as all tasks share gradient up to that point, $\nabla_{\theta} L_k = \nabla_{\theta} z \cdot \nabla_z L_k$. Thus, from now on we focus on feature-level task gradients $\nabla_z L_k$.

## 3 ROTOGRAD

In this section we introduce RotoGrad, a novel algorithm that addresses the negative transfer problem as a whole. RotoGrad consists of two building blocks which, respectively, homogenize task-gradient magnitudes and directions. Moreover, these blocks complement each other and provide convergence guarantees of the network training. Next, we detail each of these building blocks and show how they are combined towards an effective MTL learning process.

### 3.1 GRADIENT-MAGNITUDE HOMOGENIZATION

As discussed in §2, we aim to homogenize gradient magnitudes across tasks, as large magnitude disparities can lead to a subset of tasks dominating the learning process. Thus, the first goal of RotoGrad is to homogenize the magnitude of the gradients across tasks at each step of the training.

Let us denote the feature-level task gradient of the $k$-th task for the $n$-th datapoint, at iteration $t$, by $\boldsymbol{g}_{n,k} \coloneqq \nabla_z L_k(h_k(\boldsymbol{x}_n), \boldsymbol{y}_{n,k})$, and its batch versions by $\boldsymbol{G}_k^\top \coloneqq [\boldsymbol{g}_{1,k}, \boldsymbol{g}_{2,k}, \ldots, \boldsymbol{g}_{B,k}]$, where $B$ is the batch size. Then, equalizing gradient magnitudes amounts to finding weights $\omega_k$ that normalize and scale each gradient $\boldsymbol{G}_k$, that is,

$$||\omega_k \boldsymbol{G}_k|| = ||\omega_i \boldsymbol{G}_i|| \quad \forall i \iff \omega_k \boldsymbol{G}_k = \frac{C}{||\boldsymbol{G}_k||} \boldsymbol{G}_k = C \boldsymbol{U}_k \quad \forall k, \tag{1}$$

where $\boldsymbol{U}_k \coloneqq \frac{\boldsymbol{G}_k}{||\boldsymbol{G}_k||}$ denotes the normalized task gradient and $C$ is the target magnitude for all tasks. Note that, in the above expression, $C$ is a free parameter that we need to select.

In RotoGrad, we select $C$ such that all tasks converge at a similar rate. We motivate this choice by the fact that, by scaling all gradients, we change their individual step size, interfering with the convergence guarantees provided by their Lipschitz-smoothness (for an introduction to optimization see, for example, (Nesterov, 2004)). Therefore, we seek for the value of $C$ providing the best step-size for those tasks that have converged the least up to iteration $t$. Specifically, we set $C$ to be a convex combination of the task-wise gradient magnitudes, $C \coloneqq \sum_k \alpha_k ||\boldsymbol{G}_k||$, where the weights $\alpha_1, \alpha_2, \ldots, \alpha_K$ measure the relative convergence of each task and sum up to one, that is,

$$\alpha_k = \frac{||\boldsymbol{G}_k||/||\boldsymbol{G}_k^0||}{\sum_i ||\boldsymbol{G}_i||/||\boldsymbol{G}_i^0||}, \tag{2}$$

with $\boldsymbol{G}_k^0$ being the initial gradient of the $k$-th task, i.e., the gradient at iteration $t = 0$ of the training.

As a result, we obtain a (hyper)parameter-free approach that equalizes the gradient magnitude across tasks to encourage learning slow-converging tasks. Note that the resulting approach resembles Normalized Gradient Descent (NGD) (Cortés, 2006) for single-task learning, which has been proved to quickly escape saddle points during optimization (Murray et al., 2019). Thus, we expect a similar behavior for RotoGrad, where slow-converging tasks will force quick-converging tasks to escape from saddle points.

Whilst the algorithm works well in general, its simplicity also facilitates unfavorable settings. For example, in the presence of noisy tasks that do not progress; or in scenarios where, when one task improves, there is always another task that deteriorates. In Appendix A we show that, when gradients do not conflict in direction with each other (which we pursue next), following the gradient $C \sum_k \boldsymbol{U}_k$ improves all task losses for the given batch. This result, while simple, provides insights in favor of having as *desideratum* of an efficient MTL pipeline the absence of conflicting gradients.

### 3.2 GRADIENT-DIRECTION HOMOGENIZATION

In the previous subsection, we have shown that avoiding conflicting gradients may not only be necessary to avoid negative transfer, but also to ensure the stability of the training. In this section

we introduce the second building block of RotoGrad, an algorithm that homogenizes task-gradient directions. The main idea of this approach is to smoothly rotate the feature-space $z$ in order to reduce the gradient conflict between tasks—in following iterations—of the training by bringing (local) optima for different tasks closer to each other (in the parameter space). As a result, it complements the previous magnitude-scaling approach and reduces the likelihood of the training to diverge.

In order to homogenize gradients, for each task $k = 1, \ldots, K$, RotoGrad introduces a matrix $\boldsymbol{R}_k$ so that, instead of optimizing $L_k(\boldsymbol{z})$ with $\boldsymbol{z}$ being the last shared representation, we optimize an equivalent (in optimization terms, as it is a bijective mapping) loss function $L_k(\boldsymbol{R}_k \boldsymbol{z})$. As we are only interested in changing directions (not the gradient magnitudes), we choose $\boldsymbol{R}_k \in SO(d)$ to be a rotation matrix[1] leading to per-task representations $\boldsymbol{r}_k := \boldsymbol{R}_k \boldsymbol{z}$. RotoGrad thus extends the standard MTL architecture by adding task-specific rotations before each head, as depicted in Figure 2.

Unlike all other network parameters, matrices $\boldsymbol{R}_k$ do not seek to reduce their task's loss. Instead, these additional parameters are optimized to reduce the direction conflict of the gradients across tasks. To this end, for each task we optimize $\boldsymbol{R}_k$ to maximize the batch-wise cosine similarity or, equivalently, to minimize

$$\mathcal{L}_{\text{rot}}^k := -\sum_n \langle \boldsymbol{R}_k^\top \widetilde{\boldsymbol{g}}_{n,k}, \boldsymbol{v}_n \rangle, \tag{3}$$

where $\widetilde{\boldsymbol{g}}_{n,k} := \nabla_{\boldsymbol{r}_k} L_k(h_k(\boldsymbol{x}_n), \boldsymbol{y}_{n,k}))$ (which holds that $\boldsymbol{g}_{n,k} = \boldsymbol{R}_k^\top \widetilde{\boldsymbol{g}}_{n,k}$) and $\boldsymbol{v}_n$ is the target vector that we want all task gradients pointing towards. We set the target vector $\boldsymbol{v}_n$ to be the gradient we would have followed if all task gradients weighted the same, that is, $\boldsymbol{v}_n := \frac{1}{K} \sum_k \boldsymbol{u}_{n,k}$, where $\boldsymbol{u}_{n,k}$ is a row vector of the normalized batch gradient matrix $\boldsymbol{U}_k$, as defined before.

As a result, in each training step of RotoGrad we simultaneously optimize the following two problems:

$$\mathcal{N}\text{etwork:} \underset{\boldsymbol{\theta}, \{\boldsymbol{\phi}\}_k}{\text{minimize}} \sum_k \omega_k L_k \cdot, \qquad \mathcal{R}\text{otation:} \underset{\{\boldsymbol{R}_k\}_k}{\text{minimize}} \sum_k \mathcal{L}_{\text{rot}}^k \tag{4}$$

The above problem can be interpreted as a Stackelberg game: a two player-game in which *leader* and *follower* alternately make moves in order to minimize their respective losses, $L_l$ and $L_f$, and the leader knows what will be the follower's response to their moves. Such an interpretation allows us to derive simple guidelines to guarantee training convergence—that is, that the network loss does not oscillate as a result of optimizing the two different objectives in Equation 4. Specifically, following Fiez et al. (2020), we can ensure that problem 4 converges as long as the rotations' optimizer (leader) is a slow-learner compared with the network optimizer (follower). That is, as long as we make the rotations' learning rate decrease faster than that of the network, we know that RotoGrad will converge to a local optimum for both objectives. A more extensive discussion can be found in Appendix B.

### 3.3 RotoGrad: the full picture

After the two main building blocks of RotoGrad, we can now summarize the overall proposed approach in Algorithm 1. At each step, RotoGrad first homogenizes the gradient magnitudes such that there is no dominant task and the step size is set by the slow-converging tasks. Additionally, RotoGrad smoothly updates the rotation matrices—using the local information given by the task gradients—to seamlessly align task gradients in the following steps, thus reducing direction conflicts.

### 3.4 Practical considerations

In this section, we discuss the main practical considerations to account for when implementing RotoGrad and propose efficient solutions.

**Unconstrained optimization.** As previously discussed, parameters $\boldsymbol{R}_k$ are defined as rotation matrices, and thus the *Rotation* optimization in problem 4 is a constrained problem. While this would typically imply using expensive algorithms like Riemannian gradient descent (Absil et al., 2008), we can leverage recent work on manifold parametrization (Casado & Martínez-Rubio, 2019) and, instead, apply unconstrained optimization methods by automatically[2] parametrizing $\boldsymbol{R}_k$ via exponential maps on the Lie algebra of $SO(d)$.

---

[1]The special orthogonal group, $SO(d)$, denotes the set of all (proper) rotation matrices of dimension $d$.
[2]For example, Geotorch (Casado, 2019) makes this transparent to the user.

---

**Algorithm 1** Training step with RotoGrad.

---

**Input** input samples $\boldsymbol{X}$, task labels $\{\boldsymbol{Y}_k\}$, network's (RotoGrad's) learning rate $\eta$ ($\eta_{\text{roto}}$)
**Output** backbone (heads) parameters $\boldsymbol{\theta}$ ($\{\boldsymbol{\phi}_k\}$), RotoGrad's parameters $\{\boldsymbol{R}_k\}$
 1: compute shared feature $\boldsymbol{Z} = f(\boldsymbol{X}; \boldsymbol{\theta})$
 2: **for** $k = 1, 2, \ldots, K$ **do**
 3:     compute task-specific loss $L_k = \sum_n L_k(h_k(\boldsymbol{R}_k \boldsymbol{z}_n; \boldsymbol{\phi}_k), \boldsymbol{y}_{n,k})$
 4:     compute gradient of shared feature $\boldsymbol{G}_k = \nabla_{\boldsymbol{z}} L_k$
 5:     compute gradient of task-specific feature $\widetilde{\boldsymbol{G}}_k = \boldsymbol{R}_k \boldsymbol{G}_k$        ▷ Treated as constant w.r.t. $\boldsymbol{R}_k$.
 6:     compute unitary gradients $\boldsymbol{U}_k = \boldsymbol{G}_k / ||\boldsymbol{G}_k||$
 7:     compute relative task convergence $\alpha_k = ||\boldsymbol{G}_k|| / ||\boldsymbol{G}_k^0||$
 8: **end for**
 9: make $\{\alpha_k\}$ sum up to one $[\alpha_1, \alpha_2, \ldots, \alpha_K] = [\alpha_1, \alpha_2, \ldots, \alpha_K] / \sum_k \alpha_k$
10: compute shared magnitude $C = \sum_k \alpha_k ||\boldsymbol{G}_k||$
11: update backbone parameters $\boldsymbol{\theta} = \boldsymbol{\theta} - \eta \nabla_{\boldsymbol{\theta}} \boldsymbol{z} \cdot C \sum_k \boldsymbol{U}_k$
12: compute target vector $\boldsymbol{V} = \frac{1}{K} \sum_k \boldsymbol{U}_k$
13: **for** $k = 1, 2, \ldots, K$ **do**
14:     compute RotoGrad's loss $L_k^{\text{roto}} = -\sum_n \langle \boldsymbol{R}_k^\top \widetilde{\boldsymbol{g}}_{n,k}, \boldsymbol{v}_n \rangle$
15:     update RotoGrad's parameters $\boldsymbol{R}_k = \boldsymbol{R}_k - \eta_{\text{roto}} \nabla_{\boldsymbol{R}_k} L_k^{\text{roto}}$
16:     update head's parameters $\boldsymbol{\phi}_k = \boldsymbol{\phi}_k - \eta \nabla_{\boldsymbol{\phi}_k} L_k$
17: **end for**

---

**Memory efficiency and time complexity.** As we need one rotation matrix per task, we have to store $O(Kd^2)$ additional parameters. In practice, we only need $Kd(d-1)/2$ parameters due to the aforementioned parametrization and, in most cases, this amounts to a small part of the total number of parameters. Moreover, as described by Casado & Martínez-Rubio (2019), parametrizing $\boldsymbol{R}_k$ enables efficient computations compared with traditional methods, with a time complexity of $O(d^3)$ independently of the batch size. In our case, the time complexity is of $O(Kd^3)$, which scales better with respect to the number of tasks than existing methods (for example, $O(K^2d)$ for PCGrad (Yu et al., 2020)). Forward-pass caching and GPU parallelization can further reduce training time.

**Scaling-up RotoGrad.** Despite being able to efficiently compute and optimize the rotation matrix $\boldsymbol{R}_k$, in domains like computer vision, where the size $d$ of the shared representation $\boldsymbol{z}$ is large, the time complexity for updating the rotation matrix may become comparable to the one of the network updates. In those cases, we propose to only rotate a subspace of the feature space, that is, rotate only $m << d$ dimensions of $\boldsymbol{z}$. Then, we can simply apply a transformation of the form $\boldsymbol{r}_k = [\boldsymbol{R}_k \boldsymbol{z}_{1:m}, \boldsymbol{z}_{m+1:d}]$, where $\boldsymbol{z}_{a:b}$ denotes the elements of $\boldsymbol{z}$ with indexes $a, a+1, \ldots, b$. While there exist other possible solutions, such as using block-diagonal rotation matrices $\boldsymbol{R}_k$, we defer them to future work.

## 4 ILLUSTRATIVE EXAMPLES

In this section, we illustrate the behavior of RotoGrad in two synthetic scenarios, providing clean qualitative results about its effect on the optimization process. Appendix C.1 provides a detailed description of the experimental setups.

To this end, we propose two different multitask regression problems of the form

$$L(\boldsymbol{x}) = L_1(\boldsymbol{x}) + L_2(\boldsymbol{x}) = \varphi(\boldsymbol{R}_1 f(\boldsymbol{x}; \boldsymbol{\theta}), 0) + \varphi(\boldsymbol{R}_2 f(\boldsymbol{x}; \boldsymbol{\theta}), 1), \tag{5}$$

where $\varphi$ is a test function with a single global optimum whose position is parametrized by the second argument, that is, both tasks are identical (and thus related) up to a translation. We use a single input $\boldsymbol{x} \in \mathbb{R}^2$ and drop task-specific network parameters. As backbone, we take a simple network of the form $\boldsymbol{z} = \boldsymbol{W}_2 \max(\boldsymbol{W}_1 \boldsymbol{x} + \boldsymbol{b}_1, 0) + \boldsymbol{b}_2$ with $\boldsymbol{b}_1 \in \mathbb{R}^{10}, \boldsymbol{b}_2 \in \mathbb{R}^2$, and $\boldsymbol{W}_1, \boldsymbol{W}_2^\top \in \mathbb{R}^{10 \times 2}$.

For the first experiment we choose a simple (avocado-shaped) convex objective function and, for the second one, we opt for a non-convex function with several local optima and a single global optimum. Figure 1 shows the training trajectories in the presence (and absence) of RotoGrad in both experiments, depicted as level plots in the space of $\boldsymbol{z}$ and $\boldsymbol{r}_k$, respectively. We can observe that in the first experiment (Figure 1a), RotoGrad finds both optima by rotating the feature space and matching the (unique) local optima of the tasks. Similarly, the second experiment (Figure 1b) shows that, as we

have two symmetric tasks and a non-equidistant starting point, in the vanilla case the optimization is dominated by the task with an optimum closest to the starting point. RotoGrad avoids this behavior by equalizing gradients and, by aligning gradients, is able to find the optima of both functions.

## 5 RELATED WORK

Understanding and improving the interaction between tasks is one of the most fundamental problems of MTL, since any improvement in this regard would translate to all MTL systems. Consequently, several approaches to address this problem have been adopted in the literature. Among the different lines of work, the one most related to the present work is gradient homogenization.

**Gradient homogenization.** Since the problem is two-fold, there are two main lines of work. On the one hand, we have task-weighting approaches that focus on alleviating magnitude differences. Similar to us, GradNorm (Chen et al., 2018) attempts to learn all tasks at a similar rate, yet they propose to learn these weights as parameters. Instead, we provide a closed-form solution in Equation 1, and so does IMTL-G Liu et al. (2021b). However, IMTL-G scales all task gradients such that all projections of $G$ onto $G_k$ are equal. MGDA-UB (Sener & Koltun, 2018), instead, adopts an iterative method based on the Frank-Wolfe algorithm in order to find the set of weights $\{\omega_k\}$ (with $\sum_k \omega_k = 1$) such that $\sum_k \omega_k G_k$ has minimum norm. On the other hand, recent works have started to put attention on the conflicting direction problem. Maninis et al. (2019) and Sinha et al. (2018) proposed to make task gradients statistically indistinguishable via adversarial training. More recently, PCGrad (Yu et al., 2020) proposed to drop the projection of one task gradient onto another if they are in conflict, whereas GradDrop (Chen et al., 2020) randomly drops elements of the task gradients based on a sign-purity score. Contemporaneously to this work, improved versions of MGDA (Désidéri, 2012) and PCGrad have been proposed by Liu et al. (2021a) and Wang et al. (2021), respectively.

The literature also includes approaches which, while orthogonal to the gradient homogenization, are **complementary to our work** and thus could be used along with RotoGrad. Next, we provide a brief overview of them. A prominent approach for MTL is task clustering, that is, selecting which tasks should be learned together. This approach dates back to the original task-clustering algorithm (Thrun & O'Sullivan, 1996), but new work in this direction keeps coming out (Standley et al., 2020; Zamir et al., 2018; Shen et al., 2021; Fifty et al., 2021). Alternative approaches, for example, scale the loss of each task differently based on different criteria such as task uncertainty (Kendall et al., 2018), task prioritization (Guo et al., 2018), or similar loss magnitudes (Liu et al., 2021b). Moreover, while most models fall into the hard-parameter sharing umbrella, there exists other architectures in the literature. Soft-parameter sharing architectures (Ruder, 2017), for example, do not have shared parameters but instead impose some kind of shared restrictions to the entire set of parameters. An interesting approach consists in letting the model itself learn which parts of the architecture should be used for each of the tasks (Guo et al., 2020; Misra et al., 2016; Sun et al., 2020; Vandenhende et al., 2020). Other architectures, such as MTAN (Liu et al., 2019b), make use of task-specific attention to select relevant features for each task. Finally, similar issues have also been studied in other domains like meta-learning (Flennerhag et al., 2019) and continual learning (Lopez-Paz & Ranzato, 2017).

## 6 EXPERIMENTS

In this section we assess the performance of RotoGrad on a wide range of datasets and MTL architectures. First, we check the effect of the learning rates of the rotation and network updates on the stability of RotoGrad. Then, with the goal of applying RotoGrad to scenarios with large sizes of $z$, we explore the effect of rotating only a subspace of $z$. Finally, we compare our approach with competing MTL solutions in the literature, showing that RotoGrad consistently outperforms all existing methods. Refer to Appendix C for a more details on the experiments and additional results.

**Relative task improvement.** Since MTL uses different metrics for different tasks, throughout this section we group results by means of the relative task improvement, first introduced by Maninis et al. (2019). Given a task $k$, and the metrics obtained during test time by a model, $M_k$, and by a baseline model, $S_k$, which consists of $K$ networks trained on each task individually, the relative task improvement for the $k$-th task is defined as

$$\Delta_k := 100 \cdot (-1)^{l_k} \frac{M_k - S_k}{S_k},$$
(6)

where $l_k = 1$ if $M_k < S_k$ means that our model performs better than the baseline in the $k$-th task, and $l_k = 0$ otherwise. We depict our results using different statistics of $\Delta_k$ such as its mean ($\text{avg}_k \Delta_k$), maximum ($\max_k \Delta_k$), and median ($\text{med}_k \Delta_k$) across tasks.

**Statistical significance.** We highlight significant improvements according to a one-sided paired t-test ($\alpha = 0.05$), with respect to MTL with vanilla optimization (marked with † in each table).

## 6.1 TRAINING STABILITY

At the end of §3.2 we discussed that, by casting problem 4 as a Stackelberg game, we have convergence guarantees when the rotation optimizer is the slow-learner. Next, we empirically show this necessary condition.

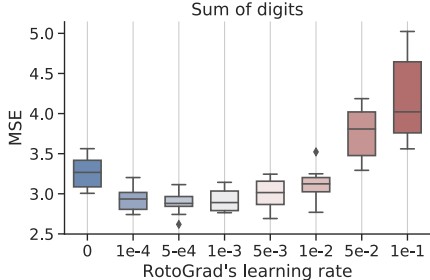

**Experimental setup.** Similar to (Sener & Koltun, 2018), we use a multitask version of MNIST (LeCun et al., 2010) where each image is composed of a left and right digit, and use as backbone a reduced version of LeNet (LeCun et al., 1998) with light-weight heads. Besides the left- and right-digit classification proposed in (Sener & Koltun, 2018), we consider three other quantities to predict: i) sum of digits; ii) parity of the digit product; and iii) number of active pixels. The idea here is to enforce all digit-related tasks to cooperate (positive transfer), while the (orthogonal) image-related task should not disrupt these learning dynamics.

Figure 3: Test error on the sum of digits task for different values of RotoGrad's learning rate on multi-MNIST.

**Results.** Figure 3 shows the effect—averaged over ten independent runs—of changing the rotations' learning rate in terms of test error in the sum task, while the rest of tasks are shown in Appendix C.2. We observe that, the bigger the learning rate is, in comparison to that of the network's parameters ($1e{-}3$), the higher and noisier the test error becomes. MSE keeps decreasing as we lower the learning rate, reaching a sweet-spot at half the network's learning rate ($5e{-}4$). For smaller values, the rotations' learning is too slow and results start to resemble those of the vanilla case, in which no rotations are applied (leftmost box in Figure 3).

## 6.2 ROTOGRAD BUILDING BLOCKS

In this section, we empirically evaluate to which extent each of the RotoGrad building blocks, that we denote *Scale Only* (§3.1) and *Rotate Only* (§3.2), contribute to the performance gain in MTL.

**Experimental setup.** We test all methods on three different tasks of NYUv2 (Couprie et al., 2013): 13-class semantic segmentation; depth estimation; and normal surfaces estimation. To speed up training, all images were resized to $288 \times 384$ resolution; and data augmentation was applied to alleviate overfitting. As MTL architecture, we use SegNet (Badrinarayanan et al., 2017) where the decoder is split into three convolutional heads. This is the same setup as that of Liu et al. (2019b).

**Results.** The three top rows of Table 1 show the performance of RotoGrad and its both components in isolation. All the methods with the same number of parameters. Compared to Vanilla optimization (4th row), *Rotate Only* improves all metrics by homogenizing gradient directions. *Scale Only* avoids overlooking the normal estimation task and improves on segmentation by homogenizing gradient magnitudes, at the expense of higher depth estimation error. Remarkably, RotoGrad exploits its scaling and rotation components to obtain the best results in semantic segmentation and depth estimation, while still achieving comparable performance in the normal estimation task.

## 6.3 SUBSPACE ROTATIONS

We now evaluate the effect of subspace rotations as described at the end of §3.4, assessing the trade-off between avoiding negative transfer and size of the subspace considered by RotoGrad.

**Experimental setup.** We test RotoGrad on a 10-task classification problem on CIFAR10 (Krizhevsky et al., 2009), using binary cross-entropy and f1-score as loss and metric, respectively, for all tasks. We use ResNet18 (He et al., 2016) without pre-training as backbone ($d = 512$), and linear layers with sigmoid activation functions as task-specific heads.

Table 1: Median (over five runs) on the NYUv2 dataset. RotoGrad obtains great performance in segmentation and depth tasks, and significantly improves the results on normal surfaces. $\Delta_S$, $\Delta_D$, and $\Delta_N$ denote the relative task improvement for each task.

| | Method | Relative improvements ↑ | | | Segmentation ↑ | | Depth ↓ | | Normal Surfaces | | | | |
| | | | | | | | | | Angle Dist. ↓ | | Within $t°$ ↑ | | |
| | | $\Delta_S$ | $\Delta_D$ | $\Delta_N$ | mIoU | Pix Acc | Abs. | Rel. | Mean | Median | 11.25 | 22.5 | 30 |
|---|---|---|---|---|---|---|---|---|---|---|---|---|---|
| | Single | 0.0 | 0.0 | 0.0 | 39.21 | 64.59 | 0.70 | 0.27 | 25.09 | 19.18 | 30.01 | 57.33 | 69.30 |
| With $\boldsymbol{R}_k$ ($m = 1024$) | Rotate Only | 3.3 | 20.5 | −6.6 | 39.63 | 66.16 | 0.53 | 0.21 | 26.12 | 20.93 | 26.85 | 53.76 | 66.50 |
| | Scale Only | −0.3 | 20.0 | −7.9 | 38.89 | 65.94 | 0.54 | 0.22 | 26.47 | 21.24 | 26.24 | 53.04 | 65.81 |
| | RotoGrad | 1.8 | 24.0 | −6.1 | 39.32 | 66.07 | 0.53 | 0.21 | 26.01 | 20.80 | 27.18 | 54.02 | 66.53 |
| | Vanilla | −2.7 | 20.6 | −25.7 | 38.05 | 64.39 | 0.54 | 0.22 | 30.02 | 26.16 | 20.02 | 43.47 | 56.87 |
| | GradDrop | −0.9 | 14.0 | −25.2 | 38.79 | 64.36 | 0.59 | 0.24 | 29.80 | 25.81 | 19.88 | 44.08 | 57.54 |
| | PCGrad | −2.7 | 20.5 | −26.3 | 37.15 | 63.44 | 0.55 | 0.22 | 30.06 | 26.18 | 19.58 | 43.51 | 56.87 |
| | MGDA-UB | −31.2 | −0.7 | 0.6 | 21.60 | 51.60 | 0.77 | 0.29 | 24.74 | 18.90 | 30.32 | 57.95 | 69.88 |
| | GradNorm | −0.6 | 19.5 | −10.5 | 37.22 | 63.61 | 0.54 | 0.22 | 26.68 | 21.67 | 25.95 | 52.16 | 64.95 |
| | IMTL-G | −0.3 | 17.6 | −7.5 | 38.38 | 64.66 | 0.54 | 0.22 | 26.38 | 21.35 | 26.56 | 52.84 | 65.69 |
| Without $\boldsymbol{R}_k$ | Vanilla† | −0.9 | 16.8 | −25.0 | 37.11 | 63.98 | 0.56 | 0.22 | 29.93 | 25.89 | 20.34 | 43.92 | 57.39 |
| | GradDrop | −0.1 | 15.7 | −27.0 | 37.51 | 63.62 | 0.59 | 0.23 | 30.15 | 26.33 | 19.32 | 43.15 | 56.59 |
| | PCGrad | −0.5 | 20.0 | −24.6 | 38.51 | 63.95 | 0.55 | 0.22 | 29.79 | 25.77 | 20.61 | 44.22 | 57.64 |
| | MGDA-UB | −32.2 | −8.2 | 1.5 | 20.75 | 51.44 | 0.73 | 0.28 | 24.70 | 18.92 | 30.57 | 57.95 | 69.99 |
| | GradNorm | 2.2 | 20.6 | −10.2 | 39.29 | 64.80 | 0.53 | 0.22 | 26.77 | 21.88 | 25.39 | 51.78 | 64.76 |
| | IMTL-G | 1.9 | 21.4 | −6.7 | 39.94 | 65.96 | 0.55 | 0.21 | 26.23 | 21.14 | 26.77 | 53.25 | 66.22 |

**Results.** Table 2 (top) shows that rotating the entire space provides the best results, and that these worsen as we decrease the size of $\boldsymbol{R}_k$. Remarkably, rotating only 128 features already outperforms vanilla with no extra per-task parameters (1st row); and rotating 256 features already yields comparable results to vanilla optimization with extra capacity (6th row) despite its larger number of (task-specific) parameters. These results can be further explained by Figure 9 in Appendix C.2, which shows a positive correlation between the size of $\boldsymbol{R}_k$ and cosine similarity.

## 6.4 METHODS COMPARISON

**Experimental setup.** In order to provide fair comparisons among methods, all experiments use identical configurations and random initializations. For all methods we performed a hyperparameter search and chose the best ones based on validation error. Unless otherwise specified, all baselines use *the same architecture (and thus, number of parameters) as RotoGrad*, taking each rotation matrix $\boldsymbol{R}_k$ as extra task-specific parameters. Further experimental details can be found in Appendix C.1, as well as extra experiments and complete results in Appendix C.2.

**NYUv2.** Table 1 shows the performance of all baselines with and without the extra capacity. RotoGrad significantly improves performance on all tasks compared with vanilla optimization, and outperforms all other baselines. Remarkably, we rotate only 1024 dimensions of $\boldsymbol{z}$ (out of a total of 7 millions) and, as a result, RotoGrad stays on par in training time with the baselines (around 4 h, Appendix C.2). We can also assert the importance of learning the matrices $\boldsymbol{R}_k$ properly by comparing in Table 1 the different baselines with and without extra capacity.

This comparison reveals that the extra parameters do not solve the negative transfer but instead amplifies biases (methods that overlook a subset of tasks, keep overlooking them) and, in the best case, provides trade-off solutions (also shown in Appendix C.2). Note, moreover, that RotoGrad (due to its *Rotate Only* component) is the only method to tackle conflicting gradient directions that manages to not overlook the normal surfaces task.

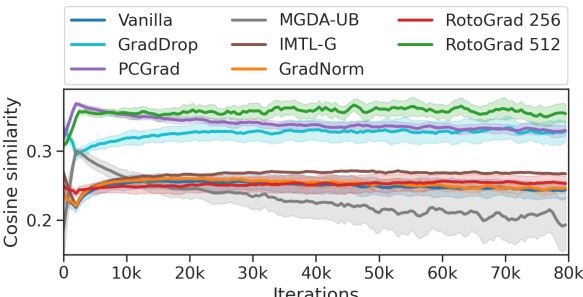

Figure 4: Similarity between task and update gradients for different methods on CIFAR10, averaged over tasks and five runs.

**CIFAR10.** We reuse the setting from §6.3 to compare different MTL baselines in terms of relative improvements (Table 2)

Table 2: (Top) Relative task improvement on CIFAR10 for RotoGrad with matrices $\boldsymbol{R}_k$ of different sizes; and (bottom) comparison with baseline methods including rotation matrices as extra task-specific parameters. Table shows median and standard deviation over five runs.

| Method | $d$ | $\mathrm{avg}_k \Delta_k \uparrow$ | $\mathrm{med}_k \Delta_k \uparrow$ | $\mathrm{max}_k \Delta_k \uparrow$ |
|---|---|---|---|---|
| Vanilla[†] | 0 | $2.58 \pm 0.54$ | $1.90 \pm 0.53$ | $11.14 \pm 3.35$ |
| RotoGrad | 64 | $2.90 \pm 0.49$ | $1.79 \pm 0.57$ | $13.16 \pm 2.40$ |
| RotoGrad | 128 | $2.97 \pm 1.08$ | $2.25 \pm 1.07$ | $12.64 \pm 3.56$ |
| RotoGrad | 256 | $3.68 \pm 0.68$ | $2.16 \pm 0.72$ | $14.01 \pm 3.22$ |
| RotoGrad | 512 | $4.48 \pm 0.99$ | $3.67 \pm 1.40$ | $15.57 \pm 3.99$ |
| | Vanilla | $3.12 \pm 0.79$ | $3.10 \pm 1.29$ | $14.23 \pm 2.86$ |
| With $\boldsymbol{R}_k$ ($m = 512$) | GradDrop | $3.54 \pm 1.10$ | $3.27 \pm 1.61$ | $13.88 \pm 2.95$ |
| | PCGrad | $3.29 \pm 0.46$ | $2.67 \pm 0.88$ | $13.44 \pm 1.86$ |
| | MGDA-UB | $0.21 \pm 0.67$ | $0.57 \pm 0.62$ | $4.78 \pm 2.15$ |
| | GradNorm | $3.21 \pm 1.04$ | $3.10 \pm 1.01$ | $10.88 \pm 4.73$ |
| | IMTL-G | $3.02 \pm 0.69$ | $1.81 \pm 0.87$ | $12.76 \pm 1.77$ |

Table 3: F1-score statistics in CelebA for two neural network architectures. Median over five different runs.

| | Method | task f1-scores (%) $\uparrow$ | | | |
|---|---|---|---|---|---|
| | | $\mathrm{min}_k$ | $\mathrm{med}_k$ | $\mathrm{avg}_k$ | $\mathrm{std}_k \downarrow$ |
| Conv. Net. with $\boldsymbol{R}_k$ ($m = 256$) | Vanilla | 4.59 | 50.28 | 56.03 | 25.65 |
| | GradDrop | 3.18 | 50.07 | 54.43 | 27.21 |
| | PCGrad | 1.44 | 53.05 | 54.72 | 27.61 |
| | GradNorm | 2.08 | 52.53 | 56.71 | 24.57 |
| | IMTL-G | 0.00 | 37.00 | 42.24 | 33.46 |
| | RotoGrad | 4.59 | 55.02 | 57.20 | 24.75 |
| ResNet18 with $\boldsymbol{R}_k$ ($m = 1536$) | Vanilla | 19.71 | 63.56 | 63.23 | 21.16 |
| | GradDrop | 12.33 | 62.40 | 62.74 | 21.74 |
| | PCGrad | 14.71 | 63.65 | 62.61 | 22.22 |
| | GradNorm | 9.05 | 60.20 | 60.78 | 22.31 |
| | IMTL-G | 17.11 | 61.68 | 60.72 | 22.80 |
| | RotoGrad | 9.96 | 63.84 | 62.81 | 21.80 |

and cosine similarity (Fig. 4), averaged over five different runs. Table 2 (bottom) shows that, similar to the NYUv2 results, both direction-aware solutions (PCGrad and GradDrop) behave similar to vanilla optimization, marginally increasing the average improvement. Unlike previous experiments, all magnitude-aware methods substantially worsen (at least) one of the statistics. In contrast, RotoGrad improves the relative task improvement across all statistics using the same number of parameters. Figure 4 shows the cosine similarity between task and update gradients, averaged over all tasks and runs (shaded areas correspond to $90\%$ confidence intervals). It is clear that RotoGrad obtains the highest cosine similarity, that other direction-aware methods also effectively align task gradients and, combined with the low cosine similarity achieved by MGDA-UB, suggests that *there exists a correlation between cosine similarity and performance*.

**CelebA.** To finalize, we test all methods in a 40-class multi-classification problem on CelebA (Liu et al., 2015) and two different settings: one using a convolutional network as backbone ($d = 512$); and another using ResNet18 (He et al., 2016) as backbone ($d = 2048$). As above, we use binary cross-entropy and f1-score as loss and metric for all tasks, thus accounting for highly imbalanced attributes. Results in Table 3 show that RotoGrad performs great in all f1-score statistics and both architectures, specially in the convolutional neural network, outperforming competing methods.

Moreover, RotoGrad achieves these results rotating $50\%$ of the shared feature $\boldsymbol{z}$ for the convolutional network, and $75\%$ for the residual network, which further demonstrates that RotoGrad can scale-up to real-world settings. We believe it is important to remark that, due to the high number of tasks, this setup is specially demanding. Results in Appendix C.2 show the performance of all baselines without the rotation matrices, demonstrating the negative effect that the extra capacity can have if not learned properly, as well as that RotoGrad stays on par with non-extended baselines in training time.

## 7 CONCLUSIONS

In this work, we have introduced RotoGrad, an algorithm that tackles negative transfer in MTL by homogenizing task gradients in terms of both magnitudes and directions. RotoGrad enforces a similar convergence rate for all tasks, while at the same time smoothly rotates the shared representation differently for each task in order to avoid conflicting gradients. As a result, RotoGrad leads to stable and accurate MTL. Our empirical results have shown the effectiveness of RotoGrad in many scenarios, staying on top of all competing methods in performance, while being on par in terms of computational complexity with those that better scale to complex networks.

We believe our work opens up interesting venues for future work. For example, it would be interesting to study alternative approaches to further scale up RotoGrad using, for example, diagonal-block or sparse rotation matrices; to rotate the feature space in application domains with structured features (e.g., channel-wise rotations in images); and to combine different methods, for example, by scaling gradients using the direction-awareness of IMTL-G and the "favor slow-learners" policy of RotoGrad.

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

# Appendices

## A  PROOFS

**Proposition A.1.** *Suppose $f_k := L_k \circ h_k$ is an infinitely differentiable real-valued function, and let us call $\boldsymbol{G}_k = \nabla_{\boldsymbol{Z}} f_k(\boldsymbol{Z})$ its derivative with respect to $\boldsymbol{Z}$, for every $k = 1, 2, \ldots, K$. If $\mathrm{cos\_sim}(\boldsymbol{G}_i, \boldsymbol{G}_j) > -1/(K-1)$ pairwise; then there exists a small-enough step size $\varepsilon > 0$ such that, for all $k$, we have that $L_k(h_k(\boldsymbol{Z} - \varepsilon \cdot C \sum_k \boldsymbol{U}_k; \boldsymbol{\phi}_k); \boldsymbol{Y}_k) < L_k(h_k(\boldsymbol{Z}; \boldsymbol{\phi}_k); \boldsymbol{Y}_k)$, where $\boldsymbol{U}_k := \boldsymbol{G}_k / ||\boldsymbol{G}_k||$ and $C \geq 0$.*

**Proof.** Since $f_k$ is infinitely differentiable, we can take the first-order Taylor expansion of $f_k$ around $\boldsymbol{Z}$, for any $k$, evaluated at $\boldsymbol{Z} - \varepsilon \boldsymbol{V}$ for a given vector $\boldsymbol{V}$:

$$f_k(\boldsymbol{Z} - \varepsilon \boldsymbol{V}) = f_k(\boldsymbol{Z}) - \varepsilon \langle \boldsymbol{G}_k, \boldsymbol{V} \rangle + o(\varepsilon). \tag{7}$$

In our case, $\boldsymbol{V} = C \sum_k \boldsymbol{U}_k$ with $C \geq 0$, then:

$$f_k(\boldsymbol{Z} - \varepsilon \boldsymbol{V}) - f_k(\boldsymbol{Z}) = -\varepsilon \cdot C ||\boldsymbol{G}_k|| \sum_i \langle \boldsymbol{U}_k, \boldsymbol{U}_i \rangle + o(\varepsilon) \tag{8}$$

$$= -\varepsilon \cdot C ||\boldsymbol{G}_k|| \left( 1 + \sum_{i \neq j} \langle \boldsymbol{U}_k, \boldsymbol{U}_i \rangle \right) + o(\varepsilon). \tag{9}$$

Since $||\boldsymbol{U}_k|| = 1$ for all $k = 1, 2, \ldots, K$, it holds that $-1 \leq \mathrm{cos\_sim}(\boldsymbol{U}_k, \boldsymbol{U}_i) = \langle \boldsymbol{U}_k, \boldsymbol{U}_i \rangle \leq 1$.

If $\mathrm{cos\_sim}(\boldsymbol{G}_k, \boldsymbol{G}_i) > -1/(K-1)$ for all $i \neq k$, then we have that $1 + \sum_{i \neq j} \langle \boldsymbol{U}_k, \boldsymbol{U}_i \rangle > 0$ and $f_k(\boldsymbol{Z} - \varepsilon \boldsymbol{V}) < f_k(\boldsymbol{V})$ for a small enough $\varepsilon > 0$.

Q.E.D.

## B  STACKELBERG GAMES AND ROTOGRAD

In game theory, a Stackelberg game (Fiez et al., 2020) is an *asymmetric game* where two players play alternately. One of the players—whose objective is to blindly minimize their loss function—is known as the follower, $\mathcal{F}$. The other player is known as the leader, $\mathcal{L}$. In contrast to the follower, the leader attempts to minimize their own loss function, but it does so with the advantage of knowing which will be the best response to their move by the follower. The problem can be written as

$$\begin{aligned} \mathcal{L}\text{eader:} \quad & \min_{x_l \in X_l} \{ \mathcal{L}(x_l, x_f) \, | \, x_f \in \operatorname*{argmin}_{y \in X_f} \mathcal{F}(x_l, y) \}, \\ \mathcal{F}\text{ollower:} \quad & \min_{x_f \in X_f} \mathcal{F}(x_l, x_f), \end{aligned} \tag{10}$$

where $x_l \in X_l$ and $x_f \in X_f$ are the actions taken by the leader and follower, respectively.

While traditionally one assumes that players make perfect alternate moves in each step of problem 10, *gradient-play Stackelberg games* assume instead that players perform simultaneous gradient updates,

$$\begin{aligned} x_l^{t+1} &= x_l^t - \alpha_l^t \, \nabla_{x_l} \mathcal{L}(x_l, x_f), \\ x_f^{t+1} &= x_f^t - \alpha_f^t \, \nabla_{x_f} \mathcal{L}(x_l, x_f), \end{aligned} \tag{11}$$

where $\alpha_l$ and $\alpha_f$ are the learning rates of the leader and follower, respectively.

An important concept in game theory is that of an equilibrium point, that is, a point in which both players are satisfied with their situation, meaning that there is no available move immediately improving any of the players' scores, so that none of the players is willing to perform additional actions/updates. Specifically, we focus on the following definition introduced by Fiez et al. (2020):

**Definition B.1** (differential Stackelberg equilibrium). A pair of points $x_l^* \in X_l$, $x_f^* \in X_f$, where $x_f^* = r(x_l^*)$ is implicitly defined by $\nabla_{x_f} \mathcal{F}(x_l^*, x_f^*) = 0$, is a differential Stackelberg equilibrium point if $\nabla_{x_l} \mathcal{L}(x_l^*, r(x_l^*)) = 0$, and $\nabla_{x_l}^2 \mathcal{L}(x_l^*, r(x_l^*))$ is positive definite.

Note that, when the players manage to reach such an equilibrium point, both of them are in a local optimum. Here, we make use of the following result, introduced by Fiez et al. (2020), to provide theoretical convergence guarantees to an equilibrium point:

**Proposition B.1.** *In the given setting, if the leader's learning rate goes to zero at a faster rate than the follower's, that is, $\alpha_l(t) = o(\alpha_f(t))$, where $\alpha_i(t)$ denotes the learning rate of player $i$ at step $t$, then they will asymptotically converge to a differential Stackelberg equilibrium point almost surely.*

In other words, as long as the follower learns faster than the leader, they will end up in a situation where both are satisfied. Even more, Fiez et al. (2020) extended this result to the finite-time case, showing that the game will end close to an equilibrium point with high probability.

As we can observe, the Stackelberg formulation in Equation (10) is really similar to RotoGrad's formulation in Equation (4). Even more, the update rule in Equation (11) is quite similar to that one shown in Algorithm 1. Therefore, it is sensible to cast RotoGrad as a Stackelberg game. One important but subtle bit about this link regards the extra information used by the leader. In our case, this extra knowledge explicitly appears in Equation 3 in the form of the follower's gradient, $\boldsymbol{g}_{i,k}$, which is the direction the follower will follow and, as it is performing first-order optimization by assumption, this gradient directly encodes the follower's response.

Thanks to the Stackelberg formulation in Equation 4 we can make use of Prop. B.1 and, thus, draw theoretical guarantees on the training stability and convergence. In other words, we can say that performing training steps as those described in Algorithm 1 will stably converge as long as the leader is asymptotically the slow learner, that is $\alpha_l^t = o(\alpha_f^t)$, where $o$ denotes the little-o notation.

In practice, however, the optimization procedure proposed by Fiez et al. (2020) requires computing the gradient of a gradient, thus incurring a significant overhead. Instead, we use Gradient Ascent-Descent (GDA), which only computes partial derivatives and enjoys similar guarantees (Jin et al., 2020), as we empirically showed in the manuscript.

## C    EXPERIMENTS

### C.1    EXPERIMENTAL SETUPS

Here, we discuss common settings across all experiments. Refer to specific sections further below for details concerning single experiments.

**Computational resources.** All experiments were performed on a shared cluster system with two NVIDIA DGX-A100. Therefore, all experiments were run with (up to) 4 cores of AMD EPYC 7742 CPUs and, for those trained on GPU (CIFAR10, CelebA, and NYUv2), a single NVIDIA A100 GPU. All experiments were restricted to 12 GB of RAM.

**Loss normalization.** Similar as in the gradient case studied in this work, magnitude differences between losses can make the model overlook some tasks. To overcome this issue, here we perform loss normalization, that is, we normalize all losses by their value at the first training iteration (so that they are all 1 in the first iteration). To account for some losses that may quickly decrease at the start, after the 20th iteration, we instead normalize losses dividing by their value at that iteration.

**Checkpoints.** For the single training of a model, we select the parameters of the model by taking those that obtained the best validation error after each training epoch. That is, after each epoch we evaluate the linearly-scalarized validation loss, $\sum_k L_k$, and use the parameters that obtained the best error during training. This can be seen as an extension of early-stopping where, instead of stopping, we keep training until reaching the maximum number of epochs hoping to keep improving.

**Baselines.** We have implemented all baselines according to the original paper descriptions, except for PCGrad, which we apply to the gradients with respect to the feature $\mathbf{z}$ (instead of the shared parameters $\boldsymbol{\theta}$, as in the original paper). Note that this is in accordance with recent works, for example Chen et al. (2020) and Liu et al. (2021b), which also use this implementation of PCGrad in the feature

space. This way, all competing methods modify gradients with respect to the same variables and, as backpropagation performs the sum of gradients at the last shared representation $z$, PCGrad can be applied to reduce conflict at that level.

**Hyperparameter tuning.** Model-specific hyperparameters were mostly selected by a combination of: i) using values described in prior works; and ii) empirical validation on the vanilla case (without any gradient-modifiers) to verify that the combinations of hyperparameters work. To select method-specific hyperparameters we performed a grid search, choosing those combinations of values that performed the best with respect to validation error.

Specifically, we took $\alpha \in \{0, 0.5, 1, 2\}$ and $\boldsymbol{R}_k \in \mathbb{R}^{m \times m}$ with $m \in \{0.25d, 0.5d, 0.75d, d\}$ (restricting ourselves to $m \leq 1500$) for RotoGrad. Regarding the learning rate of RotoGrad (GradNorm), we performed a grid search considering $\eta_{\text{roto}} \in \{0.05\eta, 0.1\eta, 0.5\eta, \eta, 2\eta\}$, where $\eta_{\text{roto}}$ and $\eta$ are the learning rates of RotoGrad (GradNorm) and the network, respectively.

**Statistical test.** For the tabular data, we highlight those results that are significantly better than those from the multitask baseline (that is, better than the vanilla MTL optimization without the $R_k$ matrices). To find these values, we run a paired one-sided Student's t-test across each method and the baseline. For those metrics for which higher is better, our null hypothesis is that the method's performance is equal or lower than the baseline, and for those for which lower is better, the null hypothesis is that the method's performance is equal or greater than the baseline. We use a significance level of $\alpha = 0.05$.

**Notation.** Along this section, we use the following to describe different architectures: `[Conv-F-C]` denotes a convolutional layer with filter size $F$ and $C$ number of channels; `[Max]` denotes a max-pool layer of filter size and stride 2, and `[Dense-H]` a dense layer with output size $H$.

### C.1.1 ILLUSTRATIVE EXAMPLES

**Losses and metrics.** Both illustrative experiments use MSE as both loss and metric. Regarding the specific form of $\varphi$ in Equation (5), the avocado-shaped experiments uses

$$\varphi((x, y), s) = (x - s)^2 + 25y^2, \tag{12}$$

while the non-convex second experiment uses

$$\varphi((x, y), s) = -\frac{\sin(3x + 4.5s)}{x + 1.5s} - \frac{\sin(3y + 4.5s)}{y + 1.5s} + |x + 1.5s| + |y + 1.5s| \tag{13}$$

**Model.** As described in the main manuscript, we use a single input $\boldsymbol{x} \in \mathbb{R}^2$ picked at random from a standard normal distribution, and drop all task-specific network parameters (that is, $h_k(\boldsymbol{r}_k) = \boldsymbol{r}_k$). As backbone, we take a simple network of the form $\boldsymbol{z} = \boldsymbol{W}_2 \max(\boldsymbol{W}_1\boldsymbol{x} + \boldsymbol{b}_1, 0) + \boldsymbol{b}_2$ with $\boldsymbol{b}_1 \in \mathbb{R}^{10}, \boldsymbol{b}_2 \in \mathbb{R}^2$, and $\boldsymbol{W}_1, \boldsymbol{W}_2^\top \in \mathbb{R}^{10 \times 2}$.

**Hyperparameters, convex-case.** We train the model for one hundred epochs. As network optimizer we use stochastic gradient descent (SGD) with a learning rate of $0.01$. For the rotations we use RAdam (Liu et al., 2019a) with a learning rate of $0.5$ (for visualization purposes we need a high learning rate, in such a simple scenario it still converges) and exponential decay with decaying factor $0.999\,99$.

**Hyperparameter, non-convex case.** For the second experiment, we train the model for 400 epochs and, once again, use SGD as the network optimizer with a learning rate of $0.015$. For the rotations, we use RAdam (Liu et al., 2019a) with a learning rate of $0.1$ and an exponential decay of $0.999\,99$.

### C.1.2 MNIST/SVHN

**Datasets.** We use two modified versions of MNIST (LeCun et al., 2010) and SVHN (Netzer et al., 2011) in which each image has two digits, one on each side of the image. In the case of MNIST, both of them are merged such that they form an overlapped image of $28 \times 28$, as shown in Figure 5a. Since SVHN contains backgrounds, we simply paste two images together without overlapping, obtaining images of size $32 \times 64$, as shown in Figure 5b. Moreover, we transform all SVHN samples to grayscale.

**Tasks, losses, and metrics.** In order to further clarify the setup used, here we describe in detail each task. Specifically, we have:

MNIST                                                         SVHN

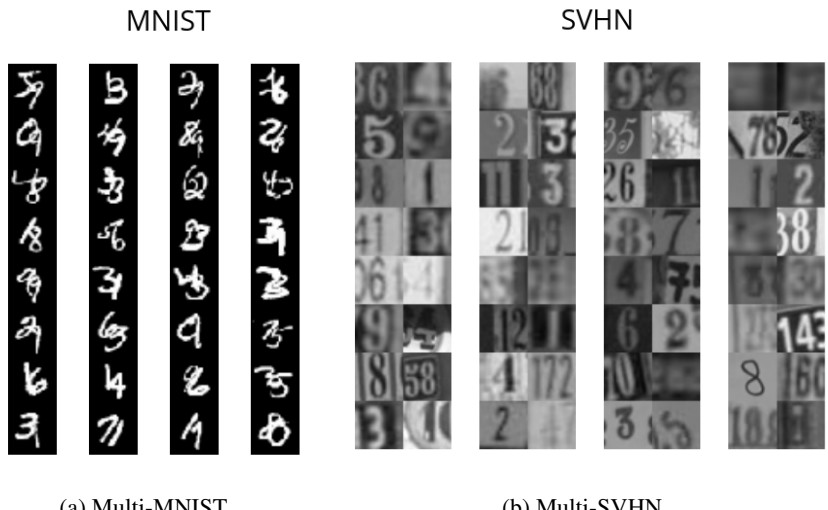

(a) Multi-MNIST.                                    (b) Multi-SVHN.

Figure 5: Samples extracted from the modified MNIST and SVHN datasets.

- **Left digit** classification. Loss: negative log-likelihood (NLL). Metric: accuracy (ACC).
- **Right digit** classification. Loss: NLL. Metric: ACC.
- **Parity** of the product of digits, that is, whether the product of both digits gives an odd number (binary-classification). Loss: binary cross entropy (BCE). Metric: f1-score (F1).
- **Sum** of both digits (regression). Loss: MSE. Metric: MSE.
- **Active pixels** in the image, that is, predict the number of pixels with values higher than 0.5, where we use pixels lying in the unit interval (regression). Loss: MSE. Metric: MSE.

**Model.** Our backbone is an adaption from the original LeNet (LeCun et al., 1998) model. Specifically, we use:

- **MNIST.** `[Conv-5-10][Max][ReLU][Conv-5-20][Max][Dense-50][ReLU][BN]`,
- **SVHN.** `[Conv-5-10][Max][ReLU][Conv-5-20][Max][Conv-5-20][Dense-50][ReLU][BN]`,

where `[BN]` refers to Batch Normalization (Ioffe & Szegedy, 2015). Depending on the type of task, we use a different head. Specifically, we use:

- **Regression.** `[Dense-50][ReLU][Dense-1]`,
- **Classification.** `[Dense-50][ReLU][Dense-10][Log-Softmax]`,
- **Binary-classification.** `[Dense-1][Sigmoid]`.

**Model hyperparameters.** For both datasets, we train the model for 300 epochs using a batch size of 1024. For the network parameters, we use RAdam (Liu et al., 2019a) with a learning rate of $1e-3$.

**Methods hyperparameters.** In Tables 4 and 5 we show the results of GradNorm with: i) MNIST with $R_k$, $\alpha = 0$; ii) MNIST without $R_k$, $\alpha = 0.5$; iii) SVHN with $R_k$, $\alpha = 1$; and iv) SVHN without $R_k$, $\alpha = 2$. We train RotoGrad with full-size rotation matrices ($m = d$). Both methods use RAdam with learning rate $5e-4$ and exponential decay of 0.9999.

### C.1.3    CIFAR10

**Dataset.** We use CIFAR10 (Krizhevsky et al., 2009) as dataset, with 40 000 instances as training data and the rest as testing data. Additionally, every time we get a sample from the dataset we: i) crop the image by a randomly selected square of size $3 \times 32 \times 32$; ii) randomly flip the image horizontally; and

iii) standardize the image channel-wise using the mean and standard deviation estimators obtained on the training data.

**Model.** We take as backbone ResNet18 (He et al., 2016) without pre-training, where we remove the last linear and pool layers. In addition, we add a Batch Normalization layer. For each task-specific head, we simply use a linear layer followed by a sigmoid function, that is, [Dense-1][Sigmoid].

**Losses and metrics.** We treat each class (out of ten) as a binary-classification task where we use BCE and F1 as loss and metric, respectively.

**Model hyperparameters.** We use a batch size of 128 and train the model for 500 epochs. For the network parameters, we use as optimizer SGD with learning rate of 0.01, Nesterov momentum of 0.9, and a weight decay of 5e−4. Additionally, we use for the network parameters a cosine learning-rate scheduler with a period of 200 iterations.

**Methods hyperparameters.** Results shown in Tables 2 and 6 use $\alpha = 0$ and $\alpha = 0.5$ for GradNorm with and without $R_k$, respectively, and we use RAdam (Liu et al., 2019a) as optimizer with learning rate 0.001 and an exponential decay factor of 0.999 95 for both GradNorm and RotoGrad.

### C.1.4 NYUv2

**Setup.** In contrast with the rest of experiments, for the NYUv2 experiments shown in §6, instead of writing our own implementation, we slightly modified the open-source code provided by Liu et al. (2019b) at https://github.com/lorenmt/mtan (commit 268c5c1). We therefore use the exact same setting as Liu et al. (2019b)—and refer to their paper and code for further details, with the addition of using data augmentation for the experiments which, although not described in the paper, is included in the repository as a command-line argument. We will provide along this work a diff file to include all gradient-modifier methods into the aforementioned code.

**Methods hyperparameters.** For the results shown in Table 1 we use GradNorm with $\alpha = 0$ and RotoGrad with rotations $R_k$ of size 1024. We use a similar optimization strategy as the rest of parameters, using Adam (Kingma & Ba, 2014) with learning rate 5e−5 (half the one of the network parameters) and where we halve this learning rate every 100 iterations.

### C.1.5 CELEBA

**Dataset.** We use CelebA (Liu et al., 2015) as dataset with usual splits. We resize each sample image so that they have size $3 \times 64 \times 64$.

**Losses and metrics.** We treat each class (out of forty) as a binary-classification task where we use BCE and F1 as loss and metric, respectively.

**ResNet model.** As with CIFAR10, we use as backbone ResNet18 (He et al., 2016) without pre-training, where we remove the last linear and pool layers. In addition, we add a Batch Normalization layer. For each task-specific head, we use a linear layer followed by a sigmoid function, that is, [Dense-1][Sigmoid].

**ResNet hyperparameters.** We use a batch size of 256 and train the model for 100 epochs. For the network parameters, we use RAdam (Liu et al., 2019a) as optimizer with learning rate 0.001 and exponential decay of 0.999 95 applied every 2400 iterations.

**Convolutional model.** For the second architecture, we use a convolutional network as backbone, [Conv-3-64][BN][Max][Conv-3-128][BN][Conv-3-128][BN][Max][Conv-3-256][BN] [Conv-3-256][BN][Max][Conv-3-512][BN][Dense-512][BN]. For the task-specific heads, we take a simple network of the form [Dense-128][BN][Dense-1][Sigmoid].

**Convolutional hyperparameters.** We use a batch size of 8 and train the model for 20 epochs. For the network parameters, we use RAdam (Liu et al., 2019a) as optimizer with learning rate 0.001 and exponential decay of 0.96 applied every 2400 iterations.

**Methods hyperparameters.** Results shown in Tables 3 and 7 use GradNorm with: i) convolutional network with $R_k$, $\alpha = 0$; ii) convolutional network without $R_k$, $\alpha = 1$; iii) residual network with $R_k$, $\alpha = 0.5$; and iv) residual network without $R_k$, $\alpha = 1$. For RotoGrad, we rotate 256 and 1536 elements of $z$ for the convolutional and residual networks. As optimizer, we use RAdam (Liu et al.,

2019a) with learning rate $5e-6$ and an exponential decay factor of $0.999\,95$ for both GradNorm and RotoGrad.

Note that for these experiments we omit MGDA-UB (Sener & Koltun, 2018) as it is computationally prohibitive in comparisons with other methods. In single-seed experiments, we however observed that it does not perform too well (specially in the convolutional network).

## C.2 Additional results

### C.2.1 Multi-MNIST and multi-SVHN

Table 4: Test performance (median and standard deviation) on two set of unrelated tasks on MNIST and SVHN, across ten different runs.

| | MNIST | | SVHN | |
| | Digits | Act Pix | Digits | Act Pix |
| Method | $\mathrm{avg}_k \Delta_k \uparrow$ | MSE $\downarrow$ | $\mathrm{avg}_k \Delta_k \uparrow$ | MSE $\downarrow$ |
|---|---|---|---|---|
| Single | $0.00 \pm 0.00$ | $0.01 \pm 0.01$ | $0.00 \pm 0.00$ | $0.17 \pm 0.06$ |
| Vanilla | $-1.43 \pm 3.24$ | $0.14 \pm 0.05$ | $4.78 \pm 0.88$ | $3.04 \pm 2.53$ |
| GradDrop | $-1.30 \pm 1.82$ | $0.16 \pm 0.04$ | $5.34 \pm 0.92$ | $2.99 \pm 2.59$ |
| PCGrad | $-1.22 \pm 2.81$ | $0.13 \pm 0.01$ | $5.01 \pm 0.65$ | $2.70 \pm 2.25$ |
| MGDA-UB | $-29.14 \pm 9.23$ | $0.06 \pm 0.00$ | $-4.36 \pm 6.72$ | $1.00 \pm 0.57$ |
| GradNorm | $0.86 \pm 1.93$ | $0.09 \pm 0.04$ | $5.24 \pm 0.89$ | $4.12 \pm 9.46$ |
| IMTL-G | $2.12 \pm 1.46$ | $0.07 \pm 0.02$ | $5.94 \pm 0.99$ | $1.70 \pm 1.05$ |
| RotoGrad | $1.55 \pm 2.22$ | $0.08 \pm 0.03$ | $6.08 \pm 0.48$ | $1.61 \pm 2.72$ |

We reuse the experimental setting from §6.1—now using the original LeNet (LeCun et al., 1998) and a multitask-version of SVHN (Netzer et al., 2011)—in order to evaluate how disruptive the orthogonal image-related task is for different methods. We can observe (Table 4) that the effect of the image-related task is more disruptive in MNIST, in which MGDA-UB utterly fails. Direction-aware methods (GradDrop and PCGrad) do not improve the vanilla results, whereas IMTL-G, GradNorm, and RotoGrad obtain the best results.

We also provide the complete results for all metrics in Table 5. In the case of MNIST, we can observe that both regression tasks tend to be quite disruptive. GradNorm, IMTL-G, and RotoGrad manage to improve over all tasks while maintaining good performance on the rest of tasks. MGDA-UB, however, focuses on the image-related task too much and overlooks other tasks. In SVHN we observe a similar behavior. This time, all methods are able to leverage positive transfer and improve their results on the parity and sum tasks, obtaining similar task improvement results. Yet, the image-related task is more disruptive than before, showing bigger differences between methods. As before, MGDA-UB completely focuses on this task, but now is able to not overlook any task while doing so. Regarding the rest of the methods, all of them improved their results with respect to the vanilla case, with RotoGrad and GradNorm obtaining the second-best results.

Table 5: Complete results (median and standard deviation) of different competing methods on MNIST/SVHN on all tasks, see Appendix C.1.2 and Appendix C.2.

| | | Method | Left digit Acc. ↑ | Right digit Acc. ↑ | Product parity f1 ↑ | Sum digits MSE ↓ | $\mathrm{avg}_k \Delta_k$ ↑ | Act. Pix. MSE ↓ |
|---|---|---|---|---|---|---|---|---|
| MNIST | | Single | $95.70 \pm 0.20$ | $94.05 \pm 0.16$ | $92.09 \pm 0.76$ | $1.90 \pm 0.10$ | $0.00 \pm 0.00$ | $0.01 \pm 0.01$ |
| | Without $\boldsymbol{R}_k$ | Vanilla[†] | $94.94 \pm 0.20$ | $93.26 \pm 0.27$ | $93.07 \pm 0.48$ | $2.10 \pm 0.17$ | $-3.26 \pm 3.12$ | $0.11 \pm 0.01$ |
| | | GradDrop | $95.33 \pm 0.39$ | $93.55 \pm 0.29$ | $93.32 \pm 0.54$ | $2.14 \pm 0.07$ | $-2.52 \pm 1.63$ | $0.13 \pm 0.02$ |
| | | PCGrad | $95.07 \pm 0.39$ | $93.28 \pm 0.18$ | $93.34 \pm 0.51$ | $2.14 \pm 0.19$ | $-3.36 \pm 3.86$ | $0.12 \pm 0.02$ |
| | | MGDA-UB | $94.46 \pm 1.04$ | $92.23 \pm 1.54$ | $83.89 \pm 1.84$ | $2.50 \pm 0.60$ | $-10.80 \pm 10.45$ | $0.06 \pm 0.02$ |
| | | GradNorm | $95.19 \pm 0.37$ | $93.70 \pm 0.31$ | $93.31 \pm 0.39$ | $2.06 \pm 28.71$ | $-1.81 \pm 37.99$ | $0.09 \pm 7.46$ |
| | | IMTL-G | $95.28 \pm 0.38$ | $93.84 \pm 0.21$ | $93.24 \pm 0.49$ | $1.91 \pm 6.61$ | $-0.01 \pm 82.48$ | $0.07 \pm 2.05$ |
| | With $\boldsymbol{R}_k$ | Vanilla | $95.13 \pm 0.20$ | $93.41 \pm 0.17$ | $93.54 \pm 0.50$ | $1.99 \pm 0.17$ | $-1.43 \pm 3.24$ | $0.14 \pm 0.05$ |
| | | GradDrop | $95.14 \pm 0.16$ | $93.47 \pm 0.12$ | $93.59 \pm 0.32$ | $2.00 \pm 0.06$ | $-1.30 \pm 1.82$ | $0.16 \pm 0.04$ |
| | | PCGrad | $95.04 \pm 0.26$ | $93.36 \pm 0.30$ | $93.49 \pm 0.30$ | $1.98 \pm 0.13$ | $-1.22 \pm 2.81$ | $0.13 \pm 0.01$ |
| | | MGDA-UB | $89.99 \pm 2.21$ | $86.76 \pm 1.18$ | $79.24 \pm 2.83$ | $3.65 \pm 0.42$ | $-29.14 \pm 9.23$ | $0.06 \pm 0.00$ |
| | | GradNorm | $95.28 \pm 0.18$ | $93.56 \pm 0.25$ | $93.56 \pm 0.57$ | $1.86 \pm 0.07$ | $0.86 \pm 1.93$ | $0.09 \pm 0.04$ |
| | | IMTL-G | $95.47 \pm 0.27$ | $93.79 \pm 0.31$ | $93.56 \pm 0.57$ | $1.73 \pm 0.09$ | $2.12 \pm 1.46$ | $0.07 \pm 0.02$ |
| | | RotoGrad | $95.45 \pm 0.19$ | $93.83 \pm 0.19$ | $93.22 \pm 0.35$ | $1.85 \pm 0.13$ | $1.55 \pm 2.22$ | $0.08 \pm 0.03$ |
| SVHN | | Single | $85.05 \pm 0.45$ | $84.58 \pm 0.24$ | $77.47 \pm 1.13$ | $5.84 \pm 0.14$ | $0.00 \pm 0.00$ | $0.17 \pm 0.06$ |
| | Without $\boldsymbol{R}_k$ | Vanilla[†] | $84.18 \pm 0.30$ | $84.18 \pm 0.38$ | $80.11 \pm 0.85$ | $4.81 \pm 0.06$ | $5.14 \pm 0.83$ | $2.75 \pm 3.17$ |
| | | GradDrop | $84.38 \pm 0.29$ | $84.48 \pm 0.41$ | $80.11 \pm 0.69$ | $4.69 \pm 0.12$ | $5.68 \pm 1.05$ | $1.91 \pm 0.86$ |
| | | PCGrad | $84.22 \pm 0.31$ | $84.23 \pm 0.21$ | $79.92 \pm 0.79$ | $4.69 \pm 0.09$ | $5.50 \pm 0.75$ | $2.26 \pm 0.85$ |
| | | MGDA-UB | $84.61 \pm 0.75$ | $84.38 \pm 0.45$ | $77.44 \pm 1.44$ | $4.47 \pm 0.18$ | $5.99 \pm 1.48$ | $0.66 \pm 0.75$ |
| | | GradNorm | $84.23 \pm 0.33$ | $84.13 \pm 0.30$ | $79.40 \pm 0.87$ | $4.92 \pm 0.07$ | $4.60 \pm 1.01$ | $4.30 \pm 2.18$ |
| | | IMTL-G | $84.60 \pm 0.45$ | $84.39 \pm 0.37$ | $79.63 \pm 1.10$ | $4.57 \pm 0.13$ | $5.81 \pm 0.85$ | $2.47 \pm 1.65$ |
| | With $\boldsymbol{R}_k$ | Vanilla | $84.11 \pm 0.48$ | $84.11 \pm 0.40$ | $79.83 \pm 0.79$ | $4.84 \pm 0.10$ | $4.78 \pm 0.88$ | $3.04 \pm 2.53$ |
| | | GradDrop | $84.23 \pm 0.35$ | $84.33 \pm 0.40$ | $80.10 \pm 0.83$ | $4.73 \pm 0.09$ | $5.34 \pm 0.92$ | $2.99 \pm 2.59$ |
| | | PCGrad | $84.21 \pm 0.21$ | $84.26 \pm 0.38$ | $79.64 \pm 0.48$ | $4.84 \pm 0.06$ | $5.01 \pm 0.65$ | $2.70 \pm 2.25$ |
| | | MGDA-UB | $77.05 \pm 5.44$ | $78.00 \pm 5.04$ | $71.76 \pm 4.32$ | $5.27 \pm 0.56$ | $-4.36 \pm 6.72$ | $1.00 \pm 0.57$ |
| | | GradNorm | $84.37 \pm 0.34$ | $84.30 \pm 0.46$ | $79.97 \pm 0.75$ | $4.72 \pm 0.13$ | $5.24 \pm 0.89$ | $4.12 \pm 9.46$ |
| | | IMTL-G | $84.23 \pm 0.34$ | $84.23 \pm 0.39$ | $79.77 \pm 1.04$ | $4.51 \pm 0.12$ | $5.94 \pm 0.99$ | $1.70 \pm 1.05$ |
| | | RotoGrad | $84.60 \pm 0.50$ | $84.44 \pm 0.45$ | $79.14 \pm 0.96$ | $4.45 \pm 0.10$ | $6.08 \pm 0.48$ | $1.61 \pm 2.72$ |

## C.2.2 ILLUSTRATIVE EXAMPLES

We complement the illustrative figures shown in Figure 1 by providing, for each example, an illustration of the effect of RotoGrad shown as an active and passive transformation. In an active transformation (Figure 6 left), points in the space are the ones modified. In our case, this means that we rotate feature $z$, obtaining $r_1$ and $r_2$, while the loss functions remain the same. In other words, for each $z$ we obtain a task-specific feature $r_k$ that optimizes its loss function. In contrast, a passive transformation (Figure 6 right) keeps the points unaltered while applying the transformation to the space itself. In our case, this translates to rotating the optimization landscape of each loss function (now we have $L_k \circ R_k$ instead of $L_K$), so that our single feature $z$ has an easier job at optimizing both tasks. In the case of RotoGrad, we can observe in both right figures that both optima lie in the same point, as we are aligning task gradients.

Besides the two regression experiments shown in §4, we include here an additional experiment where we test RotoGrad in the worst-case scenario of gradient conflict, that is, one in which task gradients are opposite to each other. To this end, we solve a 2-task binary classification problem where, as dataset, we take 1000 samples from a 2D Gaussian mixture model with two clusters; $y_{n,k} = 1$ if $x_n$ was sampled from cluster $k$; and $y_{n,k} = 0$ otherwise. We use as model a logistic regression model of the form $y_k = W_2 \max(W_1 x + b_1, 0) + b_2$ with $b_1 \in \mathbb{R}^2, b_2 \in \mathbb{R}, W_1 \in \mathbb{R}^{2\times2}$, and $W_2 \in \mathbb{R}^{1\times2}$. Because rotations in 1D are ill-posed (there is a unique proper rotation), here we add task parameters to increase the dimensionality of $z$ and make *all parameters shared*, so that there is still no

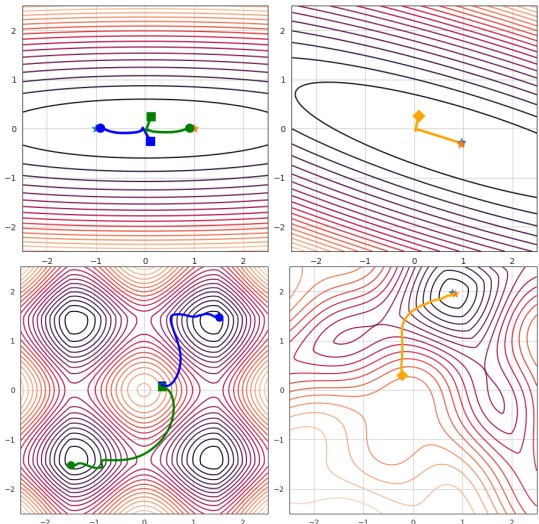

Figure 6: Level plots showing the illustrative examples of Figure 1 for RotoGrad. Top: Convex case. Bottom: Non-convex case. Left: Active transformation (trajectories of $r_k$ and the level plot of $L_1 + L_2$. Right: Passive transformation (trajectory of $z$ and level plot of $(L_1 \circ R_1) + (L_2 \circ R_2)$).

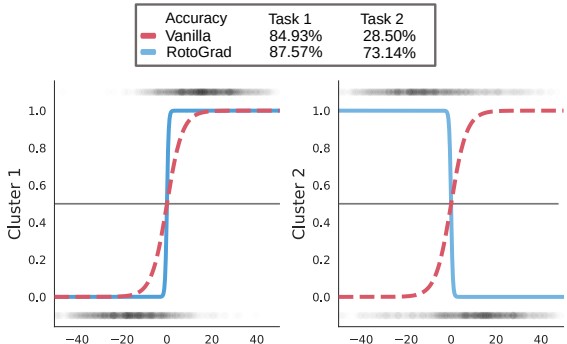

Figure 7: Logistic regression for opposite classification tasks. Test data is plotted scattered as gray dots. RotoGrad learns both opposite rotations $R_1 = R_2^\top$.

task-specific parameters. To avoid a complete conflict where $\nabla_z L_1 + \nabla_z L_2 = 0$, we randomly flip the labels for the second tasks with $5\,\%$ probability. Figure 7 shows that, in this extreme scenario, RotoGrad is able to learn both tasks by aligning gradients, that is, by learning that one rotation is the inverse of the other $R_1 = R_2^\top$.

## C.2.3 TRAINING STABILITY

While we showed in §6.1 only the results for the sum-of-digits task as they were nice and clear, here we show in Figure 8 the results of those same experiments in §6.1 for all the different tasks. The same discussion from the main manuscript can be carried out for all metrics. Additionally, we can observe that the vanilla case (learning rate zero) completely overlooks the image-related task (*Active pixels*) while performing the best in the parity task.

Additionally, let us clarify what we mean here with stability, as in the main manuscript we mainly talked about convergence guarantees. In these experiments we measure the convergence guarantees

of the experiments in terms of 'training stability', meaning the variance of the obtained results across different runs. The intuition here is that, since the model does not converge, we should expect some wriggling learning curves during training and, as we take the model with the best validation error, the individual task metrics should have bigger variance (that is, less stability) across runs.

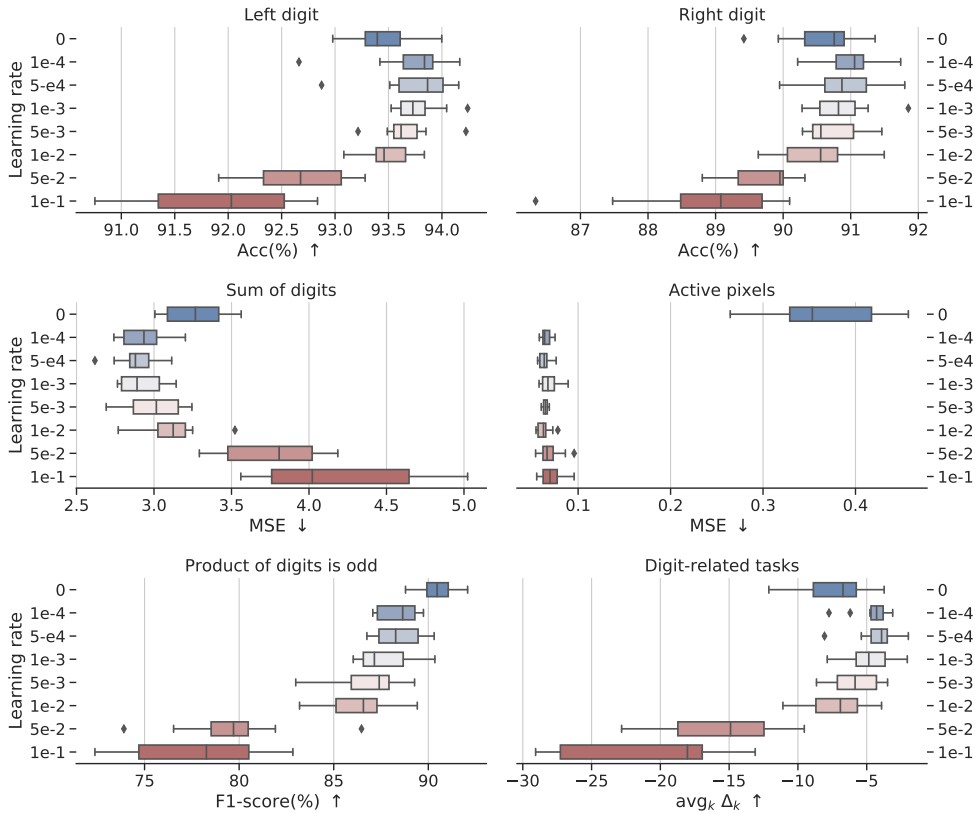

Figure 8: RotoGrad's performance on all tasks for the experiments in §6.1 for all metrics. We can observe training instabilities/stiffness on all tasks as we highly increase/decrease RotoGrad's learning rate, as discussed in the main manuscript.

### C.2.4  CIFAR10 AND CELEBA

For the sake of completeness, we present in Table 6 and Table 7 the same tables as in §6, but with more statistics of the results. For CIFAR10, we now included in Table 6 the minimum task improvement across tasks and, while noisier, we can still observe that RotoGrad also improve this statistic. The standard deviation of the task improvement across tasks is, however, not too informative. We also include in Figure 9 the cosine similarity plots for the different subspace rotations from §6.3, showing a clear positive correlation between the cosine similarity and the size of the considered subspace. While the cosine similarities look low, we want to remark that we are computing the cosine similarity of a huge space, and we only align a subspace of it. If we, instead, showed the cosine similarity with respect to each specific subspace, the cosine similarity should look similar to that of RotoGrad 512. In the case of CelebA, we added in Table 7 the maximum f1-score across tasks and, similar to the last case, it is not too informative, as all methods achieve almost perfect f1-score in one of the classes. We also include the training times for some baselines, showing that RotoGrad stays on par with them.

### C.2.5  NYUV2

Complementing the results shown in §6, we show in Table 8 the results obtained combining RotoGrad with all other existing methods (rows within RotoGrad +), for gradient scaling methods we only apply the rotation part of RotoGrad. Results show that RotoGrad helps improve/balance all other

Table 6: Complete task-improvement statistics in CIFAR10 for all competing methods and RotoGrad with different dimensionality for $\boldsymbol{R}_k$, see §6.

| Method | $d$ | $\min_k \Delta_k \uparrow$ | $\mathrm{med}_k \Delta_k \uparrow$ | $\mathrm{avg}_k \Delta_k \uparrow$ | $\mathrm{std}_k \Delta_k \downarrow$ | $\max_k \Delta_k \uparrow$ |
|---|---|---|---|---|---|---|
| Vanilla[†] | 0 | $-0.81 \pm 0.37$ | $1.90 \pm 0.53$ | $2.58 \pm 0.54$ | $3.38 \pm 0.94$ | $11.14 \pm 3.35$ |
| RotoGrad | 64 | $-1.70 \pm 0.81$ | $1.79 \pm 0.57$ | $2.90 \pm 0.49$ | $3.98 \pm 0.62$ | $13.16 \pm 2.40$ |
| RotoGrad | 128 | $-1.12 \pm 0.36$ | $2.25 \pm 1.07$ | $2.97 \pm 1.08$ | $3.84 \pm 0.87$ | $12.64 \pm 3.56$ |
| RotoGrad | 256 | $0.17 \pm 1.01$ | $2.16 \pm 0.72$ | $3.68 \pm 0.68$ | $3.83 \pm 0.74$ | $14.01 \pm 3.22$ |
| RotoGrad | 512 | $-0.43 \pm 0.76$ | $3.67 \pm 1.40$ | $4.48 \pm 0.99$ | $4.23 \pm 0.82$ | $15.57 \pm 3.99$ |
| *Without $\boldsymbol{R}_k$* Vanilla[†] | | $-0.81 \pm 0.37$ | $1.90 \pm 0.53$ | $2.58 \pm 0.54$ | $3.38 \pm 0.94$ | $11.14 \pm 3.35$ |
| GradDrop | | $-0.73 \pm 0.33$ | $2.80 \pm 0.20$ | $3.41 \pm 0.45$ | $4.08 \pm 0.34$ | $13.58 \pm 1.50$ |
| PCGrad | | $-1.52 \pm 0.98$ | $1.95 \pm 0.87$ | $2.86 \pm 0.81$ | $3.74 \pm 0.69$ | $12.01 \pm 3.19$ |
| MGDA-UB | | $-7.27 \pm 1.36$ | $-1.21 \pm 0.74$ | $-1.75 \pm 0.43$ | $3.24 \pm 0.55$ | $3.67 \pm 0.98$ |
| GradNorm | | $-0.35 \pm 0.59$ | $2.45 \pm 0.66$ | $3.23 \pm 0.35$ | $4.02 \pm 0.33$ | $14.25 \pm 1.35$ |
| IMTL-G | | $-0.39 \pm 0.82$ | $1.97 \pm 0.29$ | $2.73 \pm 0.27$ | $3.25 \pm 0.75$ | $10.20 \pm 2.98$ |
| *With $\boldsymbol{R}_k$ ($d = 512$)* Vanilla | | $-0.85 \pm 0.58$ | $3.10 \pm 1.29$ | $3.12 \pm 0.79$ | $4.05 \pm 0.56$ | $14.23 \pm 2.86$ |
| GradDrop | | $-1.49 \pm 0.78$ | $3.27 \pm 1.61$ | $3.54 \pm 1.10$ | $4.11 \pm 0.56$ | $13.88 \pm 2.95$ |
| PCGrad | | $-1.44 \pm 0.58$ | $2.67 \pm 0.88$ | $3.29 \pm 0.46$ | $3.90 \pm 0.37$ | $13.44 \pm 1.86$ |
| MGDA-UB | | $-3.59 \pm 1.48$ | $0.57 \pm 0.62$ | $0.21 \pm 0.67$ | $2.44 \pm 0.52$ | $4.78 \pm 2.15$ |
| GradNorm | | $-0.79 \pm 1.28$ | $3.10 \pm 1.01$ | $3.21 \pm 1.04$ | $3.41 \pm 0.86$ | $10.88 \pm 4.73$ |
| IMTL-G | | $-1.29 \pm 0.52$ | $1.81 \pm 0.87$ | $3.02 \pm 0.69$ | $3.81 \pm 0.21$ | $12.76 \pm 1.77$ |
| RotoGrad | | $-0.43 \pm 0.76$ | $3.67 \pm 1.40$ | $4.48 \pm 0.99$ | $4.23 \pm 0.82$ | $15.57 \pm 3.99$ |

Table 7: Complete f1-score statistics and training hours in CelebA for all competing methods and two different architectures/settings (median over five runs), see §6. For the convolutional network we use $m = 256$, and $m = 1536$ for the residual network.

| | Method | Convolutional ($d = 512$) task f1-scores (%) $\uparrow$ | | | | | | ResNet18 ($d = 2048$) task f1-scores (%) $\uparrow$ | | | | | |
|---|---|---|---|---|---|---|---|---|---|---|---|---|---|
| | | $\min_k$ | $\mathrm{med}_k$ | $\mathrm{avg}_k$ | $\mathrm{std}_k \downarrow$ | $\max_k$ | Hours | $\min_k$ | $\mathrm{med}_k$ | $\mathrm{avg}_k$ | $\mathrm{std}_k \downarrow$ | $\max_k$ | Hours |
| *Without $\boldsymbol{R}_k$* | Vanilla[†] | 1.62 | 53.39 | 58.49 | 24.26 | 96.97 | 7.62 | 15.45 | 63.04 | 62.85 | 22.09 | 96.58 | 1.49 |
| | GradDrop | 2.63 | 52.32 | 57.33 | 25.27 | 96.72 | 8.53 | 13.31 | 64.37 | 63.95 | 20.93 | 96.59 | 1.60 |
| | PCGrad | 2.69 | 54.60 | 56.87 | 25.75 | 97.04 | 34.05 | 13.61 | 62.45 | 62.74 | 21.60 | 96.64 | 5.75 |
| | GradNorm | 2.17 | 52.98 | 56.91 | 24.72 | 96.84 | 20.93 | 17.42 | 62.49 | 62.62 | 21.93 | 96.55 | 3.61 |
| | IMTL-G | 0.00 | 14.81 | 31.90 | 33.58 | 93.31 | 9.46 | 9.87 | 62.22 | 62.03 | 22.47 | 96.51 | 1.73 |
| *With $\boldsymbol{R}_k$* | Vanilla | 4.24 | 49.85 | 55.33 | 26.03 | 96.88 | 16.29 | 19.71 | 63.56 | 63.23 | 21.16 | 96.55 | 9.33 |
| | GradDrop | 3.18 | 50.07 | 54.43 | 27.21 | 96.80 | 17.20 | 12.33 | 62.40 | 62.74 | 21.74 | 96.65 | 9.41 |
| | PCGrad | 1.44 | 53.05 | 54.72 | 27.61 | 96.90 | 41.79 | 14.71 | 63.65 | 62.61 | 22.22 | 96.59 | 13.72 |
| | GradNorm | 2.08 | 52.53 | 56.71 | 24.57 | 96.96 | 30.02 | 9.05 | 60.20 | 60.78 | 22.31 | 96.38 | 11.36 |
| | IMTL-G | 0.00 | 37.00 | 42.24 | 33.46 | 94.34 | 18.05 | 17.11 | 61.68 | 60.72 | 22.80 | 96.44 | 9.52 |
| | RotoGrad | 4.59 | 55.02 | 57.20 | 24.75 | 96.79 | 27.20 | 9.96 | 63.84 | 62.81 | 21.80 | 96.45 | 6.68 |

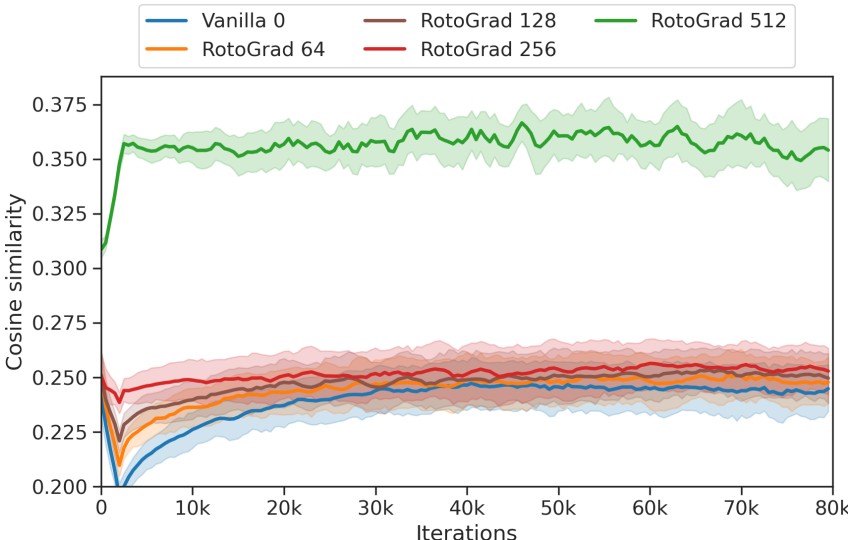

Figure 9: Cosine similarity between the task gradients and the update gradient on CIFAR10. Results are averaged over tasks and five runs, showing $90\%$ confidence intervals (shaded areas).

Table 8: Results for different methods on the NYUv2 dataset with a SegNet model. RotoGrad obtains the best performance in segmentation and depth tasks on all metrics, while significantly improving the results on normal surfaces with respect to the vanilla case.

| | Method | Relative improvements ↑ ΔS | ΔD | ΔN | Segmentation ↑ mIoU | Pix Acc | Depth ↓ Abs. | Rel. | Normal Surfaces Angle Dist. ↓ Mean | Median | Within $t°$ ↑ 11.25 | 22.5 | 30 | Time ↓ h |
|---|---|---|---|---|---|---|---|---|---|---|---|---|---|---|
| | Single | 0.0 | 0.0 | 0.0 | 39.21 | 64.59 | 0.70 | 0.27 | 25.09 | 19.18 | 30.01 | 57.33 | 69.30 | 8.90 |
| RotoGrad + | GradDrop | 1.2 | 12.6 | −7.5 | 40.26 | 65.63 | 0.63 | 0.24 | 26.33 | 21.08 | 26.47 | 53.38 | 66.05 | 3.94 |
| | PCGrad | −0.0 | 19.7 | −8.3 | 39.08 | 64.68 | 0.54 | 0.21 | 26.41 | 21.29 | 26.13 | 52.99 | 65.72 | 3.89 |
| | MGDA-UB | 2.5 | 23.2 | −8.1 | 39.32 | 65.48 | 0.54 | 0.21 | 26.43 | 21.22 | 26.16 | 53.16 | 66.07 | 3.85 |
| | GradNorm | 1.1 | 21.4 | −7.7 | 39.08 | 65.43 | 0.54 | 0.21 | 26.44 | 21.42 | 26.17 | 52.59 | 65.52 | 3.84 |
| | IMTL-G | 1.7 | 21.2 | −6.9 | 40.13 | 65.17 | 0.55 | 0.21 | 26.20 | 21.06 | 26.69 | 53.39 | 66.04 | 3.96 |
| With $\boldsymbol{R}_k$ ($m = 1024$) | Rotate Only | 3.3 | 20.5 | −6.6 | 39.63 | 66.16 | 0.53 | 0.21 | 26.12 | 20.93 | 26.85 | 53.76 | 66.50 | 3.82 |
| | Scale Only | −0.3 | 20.0 | −7.9 | 38.89 | 65.94 | 0.54 | 0.22 | 26.47 | 21.24 | 26.24 | 53.04 | 65.81 | 3.87 |
| | RotoGrad | 1.8 | 24.0 | −6.1 | 39.32 | 66.07 | 0.53 | 0.21 | 26.01 | 20.80 | 27.18 | 54.02 | 66.53 | 3.83 |
| | Vanilla | −2.7 | 20.6 | −25.7 | 38.05 | 64.39 | 0.54 | 0.22 | 30.02 | 26.16 | 20.02 | 43.47 | 56.87 | 3.81 |
| | GradDrop | −0.9 | 14.0 | −25.2 | 38.79 | 64.36 | 0.59 | 0.24 | 29.80 | 25.81 | 19.88 | 44.08 | 57.54 | 4.01 |
| | PCGrad | −2.7 | 20.5 | −26.3 | 37.15 | 63.44 | 0.55 | 0.22 | 30.06 | 26.18 | 19.58 | 43.51 | 56.87 | 3.89 |
| | MGDA-UB | −31.2 | −0.7 | 0.6 | 21.60 | 51.60 | 0.77 | 0.29 | 24.74 | 18.90 | 30.32 | 57.95 | 69.88 | 3.85 |
| | GradNorm | −0.6 | 19.5 | −10.5 | 37.22 | 63.61 | 0.54 | 0.22 | 26.68 | 21.67 | 25.95 | 52.16 | 64.95 | 3.85 |
| | IMTL-G | −0.3 | 17.6 | −7.5 | 38.38 | 64.66 | 0.54 | 0.22 | 26.38 | 21.35 | 26.56 | 52.84 | 65.69 | 3.99 |
| Without $\boldsymbol{R}_k$ | Vanilla† | −0.9 | 16.8 | −25.0 | 37.11 | 63.98 | 0.56 | 0.22 | 29.93 | 25.89 | 20.34 | 43.92 | 57.39 | 3.46 |
| | GradDrop | −0.1 | 15.7 | −27.0 | 37.51 | 63.62 | 0.59 | 0.23 | 30.15 | 26.33 | 19.32 | 43.15 | 56.59 | 3.55 |
| | PCGrad | −0.5 | 20.0 | −24.6 | 38.51 | 63.95 | 0.55 | 0.22 | 29.79 | 25.77 | 20.61 | 44.22 | 57.64 | 3.51 |
| | MGDA-UB | −32.2 | −8.2 | 1.5 | 20.75 | 51.44 | 0.73 | 0.28 | 24.70 | 18.92 | 30.57 | 57.95 | 69.99 | 3.52 |
| | GradNorm | 2.2 | 20.6 | −10.2 | 39.29 | 64.80 | 0.53 | 0.22 | 26.77 | 21.88 | 25.39 | 51.78 | 64.76 | 3.50 |
| | IMTL-G | 1.9 | 21.4 | −6.7 | 39.94 | 65.96 | 0.55 | 0.21 | 26.23 | 21.14 | 26.77 | 53.25 | 66.22 | 3.61 |

methods, which is specially true for those methods that heavily overlook some tasks. Specifically, MGDA-UB stops overlooking the semantic segmentation and depth estimation tasks, while PCGrad and GradDrop stop completely overlooking the surface normal loss. Note that we also show in Table 8 the training times of each method, and RotoGrad stays on par with non-extended methods in training time. As mentioned in Appendix C.1, due to cluster overload, some times were deceivingly high (specifically those baselines with $\boldsymbol{R}_k$) as we had to run them on different machines, and were omitted to avoid confusion.

