# OpenReview forum: "RotoGrad: Gradient Homogenization in Multitask Learning"
_ICLR.cc/2022/Conference — ICLR 2022 Spotlight_

### Official Review · Reviewer_2soJ · 2021-10-28

**Correctness:** 4
**Technical Novelty And Significance:** 3
**Empirical Novelty And Significance:** 3
**Recommendation:** 8
**Confidence:** 4

**Main Review:**

Early recent work (2015-2018) in multi-task learning focused on dynamic task reweighing methods (i.e. how to change the loss scale throughout training), but it seems that the community has shifted to focusing on direct gradient modifications (2020-2021) with a plethora of methods being developed that modify both the scale and direction of the gradient update on the shared parameters (PCGrad, GradDrop, GradVac, IMTL, CAGrad, etc.). With at least 5 somewhat conceptually similar methods having been released in the past year, it's more important than before for new approaches in this space to offer a new perspective or approach this challenge from a different direction.

It is my opinion this work meets the requisite threshold. The concept of applying (and learning) a geometric rotation to align the optimization trajectories is novel; although, the concept of gradient-magnitude homogenization has been well-studied, with the presentation offered by this work feeling very reminiscent of GradNorm. Returning to the former point, I have not seen a rotation transformation in SO(d) applied to MTL -- the closest analogue are modulation modules which filter/modify information in either the forward and/or backward pass -- and I think the dissemination of this perspective may lead to other interesting and powerful work.

Nevertheless, the current conceptualization of rotation to maximize cosine similarity may have some problems. Two identical tasks with identical initialization will have perfect alignment, and results in an effective doubling of the learning rate for the shared parameters, but does not lead to the benefits that may be conveyed from multi-task learning models (i.e. inductive bias on different hypothesis classes, regularization, etc.). Moreover, the (hyper)parameter free approach which enforces a strict constraint on equal norm among all gradients seems highly prohibitive (and perhaps undesirable since certain tasks should naturally train at different rates?). That said, many of the methods I mention in this first paragraph adopt a similar perspective with regards to a correlation between positive cosine similarity and modeling performance, and one can invert the question to posit that two tasks with perfectly opposite gradients will not even begin to train.

Strengths:

1. The approach is well motivated with illustrated examples and intuition.
2. Extremely well written and polished paper.
3. Empirical methodology is sound. It appears as if all methods are fairly compared and assessed.
4. Experimental results are compelling.

Weaknesses:

1. I don't see any significant flaws in this work.

Nitpicks:

1. *to the shard feature z* => to the shared feature representation z.
2. The results in the table are a bit difficult to read as nothing stands out. Perhaps rerun the NYUv2 experiments 3-5 times and bold significant results?
3. *While recent work has acknowledged that negative transfer is a two-fold problem, existing approaches fall short as they only focus on either homogenizing the gradient magnitude across tasks; or greedily change the gradient directions, overlooking future conflicts.* => This sentence is grammatically off. Consider breaking it into two smaller sentences.
4.  *Previous works have tracked down this issue to the disparities in gradient magnitudes and directions across tasks, when optimizing the shared network parameters.* => remove the comma or bring the dependent clause (when optimizing the shared network parameters) to the beginning of the sentence (i.e. When optimizing the shared network parameters, prior wok has indicated discrepancies in gradient magnitude among tasks may inhibit model performance or something like this).
5. Several *very* recent papers have been overlooked that should probably be included at different points in the related work section.

[1] Gradient Vaccine: Investigating and Improving Multi-task Optimization in Massively Multilingual Models, ICLR 2021 (spotlight)

[2] Conflict-Averse Gradient Descent for Multi-task learning, NeurIPS 2021

[3] Efficiently Identifying Task Groupings for Multi-Task Learning, NeurIPS 2021 (spotlight)

[4] Variational Multi-Task Learning with Gumbel-Softmax Priors, NeurIPS 2021

**Summary Of The Paper:**

This work offers an approach to improve the training dynamics of multi-task neural networks by making two assumptions:

1. Magnitude differences among task gradients inhibit convergence.
2. Directional differences (often termed "conflicting gradients") in the gradients among tasks inhibit convergence.

It adapts the optimization process by:

1. Ensuring all gradients have equal norm.
2. Introducing a geometric rotation in SO(d) transformation after the final shared feature map and before the task-specific layers.

The intuition underpinning this method is supported with several illustrative (toy) examples as well as evaluation on MNIST, CelebA, CIFAR-10, and NYUv2.

**Summary Of The Review:**

I think this is a good paper. It is clearly written, the empirical evaluation is fair, and presents a novel perspective (rotation) to reduce inter-task conflict in multi-task learning paradigms.

I recommend it for acceptance and have no additional or follow-up questions.

---

> ### Author Response · Authors · 2021-11-12
> **Reply to Reviewer Reviewer 2soJ**
>
> Dear Reviewer 2soJ,
>
> We would like to first thank you for the nice and thought-provoking review. It is a great summary of the past/current trends for palliating negative transfer. Specially, we agree on the necessity of providing new perspectives to tackle the problem. We sincerely hope that RotoGrad contributes to establish new research directions on negative transfer.
>
> We agree with the reviewer on the shortcomings of making all task gradients identical, and we feel that more work needs to be put to fully understand its implications. With regard to RotoGrad, our intuition is that such problems might arise only in trivial cases. Since the optimization landscapes across tasks are likely different, and our solution is restricted to rotations, RotoGrad will not lead to perfect alignment at every optimization step. RotoGrad only attempts to slowly align local optima, and thus, there should still exist useful information transfer across tasks to update the shared parameters.
>
> Our goal in this work is that all tasks are learnt at similar rates. We are aware of the potential shortcomings of our approach, and we briefly discuss them at the end of §3.1. Moreover, we are planning as future work to investigate, either theoretically or empirically, to which extent enforcing perfect gradient alignment may be harmful in some settings (for example, when tasks are independent of each other, and thus positive transfer is not possible). This potential issue is precisely the reason why we decided to add the (non-related) active-pixel task on MNIST/SVHN dataset (§C2.1), for which RotoGrad provides robust results.
>
> Nitpicks:
>
> - We will address the grammatical issues in the new revision (points 1, 3, 4)
> - We will happily include all these references in the related work section, and we thank the reviewer for pointing them out. Please note, however, that [2,3,4] were under review at the time of submitting this work.
> - At submission time, we ran a single seed for the NYUv2 experiments, as they are computationally demanding. But we agree with the reviewer's comment, and we will run more seeds. While we may not have them at the end of the rebuttal period, we will make sure the results are updated for the revised (and hopefully camera-ready) version.

---

### Official Review · Reviewer_gsqy · 2021-11-01

**Correctness:** 4
**Technical Novelty And Significance:** 3
**Empirical Novelty And Significance:** 3
**Recommendation:** 8
**Confidence:** 3

**Main Review:**

- **Questions:**
    1. Why do you need to specifically enforce $R_k$ to be a rotation matrix? Why it cannot be free to be a (non)-linear transformation? Have you tried experiments in this direction? Would that make sense?
    2. Regarding the comparison with other methods: From the supplementary, it looks that the PCGrad projection has been computed differently (wrt features rather than weights), can you elaborate more on this difference?
    3. By looking at the PCgrad paper (which seems to be your main "competitor") I found that they apply pcgrad in combination with other multitask methods, e.g. MTAN. Can ROTOGRAD work together with other methods? Did you try these experiments?
    4. It would be great to go beyond supervised learning experiments and include multitask RL experiments in your analysis.
- **Minors**
    - I suggest guiding the reader a bit more on table1, by highlighting the best and worst results as you do in the supplementary.


**Summary Of The Paper:**

- This paper proposes a new way of dealing with negative transfer or catastrophic interference in multitask learning. The main idea is to avoid conflicting gradients by making them homogeneous both in terms of magnitude and direction. Gradient Magnitude are normalized by taking into account the convergence rate of all tasks. Directions are homogenised by smoothly rotating the feature-space introducing rotation matrices $R_k \in SO(d)$ parametrized via exponential maps on the Lie algebra of $SO(d)$.
- Rotation matrices are optimized to reduce the direction conflicts by maximizing the cosine similarity within the the batch, while the resto of the network is trained to minimize the multitask loss. The training algorithm result in an alternated optimization procedure that can be interpreted as a Stackelberg game with a leader and a follower alternatively minimizing their own losses.


**Summary Of The Review:**

The paper is solid and it looks has received several iterations. It is a nice contribution to the multi-task learning field, although I think there are still some questions to be addressed first. I'm leaning towards acceptance and I'll be more than happy to increase my rate after the rebuttal and discussion with other reviewers.

---

> ### Author Response · Authors · 2021-11-12
> **Reply to Reviewer gsqy**
>
> Dear Reviewer gsqy,
>
> Thank you for your thorough review and useful comments. We hope the following responses help you clarify the questions you had during the review process. If that were not the case, please do not hesitate to continue with any follow-up question.
>
> Questions:
>
> 1. _Replacing rotation matrix by a non-linear transformation_. Initially, we tried other approaches, such as using a free matrix $A$. However, we encountered numerical issues related with scaling the feature $z$ in the forward pass.
> In contrast, RotoGrad only scales the gradients in the backward pass, while the rotation $R_k$ allows us to homogenize the gradient _directions_. As a result, we can efficiently compute and optimize RotoGrad's loss (see Eq.(3) and the line below).
> Yet, while it might be possible to apply other transformations, we currently fail to see how. We will further investigate that in future work.
>
> 2. Regarding the details of PCGrad in the feature space $z$, we rely on the same implementation as previous works (see, for example, [1] and [2]). We do so for fair comparison across methods, as all baselines modify the gradients wrt the shared feature $z$. Note that during backpropagation, the actual sum of gradients is performed in $z$, and thus PCGrad can be applied there to reduce the conflict at that level. We will clarify the details of this implementation in the Appendix.
>
> 3. We believe that RotoGrad can be combined with most negative-transfer solutions (see, for example, Table 8 in §C2.5). For architectural solutions, such as MTAN, RotoGrad should in principle be applied directly to the shared parameters (if computational complexity permits), or to the shared features $z$, if identified (it is not as clear in this type of architectures). We will add this discussion to the conclusions.
>
> 4. While the focus on this work is supervised multitask learning, we agree that it would be indeed interesting to apply RotoGrad to reinforcement learning experiments. We defer this application to future work.
>
>
> Minors:
>
> - We really appreciate the suggestion. We will update the draft to highlight the most important results on Table 1 (similar to what we do on Table 5).
>
> [1] Just Pick a Sign: Optimizing Deep Multitask Models with Gradient Sign Dropout - NeurIPS 2020
>
> [2] Towards Impartial Multi-task Learning - ICLR 2021

---

> > ### Comment · Reviewer_gsqy · 2021-11-29
> > **final rate**
> >
> > Thank you for your answer. I've increased my score to 8 as I think the paper should be accepted as a poster for the conference.

---

### Official Review · Reviewer_cExt · 2021-11-05

**Correctness:** 3
**Technical Novelty And Significance:** 4
**Empirical Novelty And Significance:** 4
**Recommendation:** 8
**Confidence:** 4

**Main Review:**

**Strengths:**
- The paper is well written and well structured.
- The paper tackles an important problem and proposes an intuitive approach for alleviating the problem. The motivation for the method is clear.
- The strong experimental section presents a clear improvement over previous MTL approaches.

**Weaknesses:**
- It would be useful to fully observe and understand the tradeoff between $m$ and runtime, as the scalability of the approach for large values of $d$ is limited.
- It appears the proposed update step may increase the loss (objective) for some tasks. While intuitive, it’s not clear to me if the average gradient is the right choice for the target vector ($v_n$). Have you experimented with other choices for $v_n$, for example, the common descent direction in MGDA?
- Missing citation to relevant works: [1]  propose a method to handle conflicting gradients in MTL. [2] theoretically discuss challenges of MTL optimization. Specifically, while different significantly than the proposed method, their covariance alignment approach appears to relate to the rotation matrices in the proposed method (although being applied to the input instead of the shared feature space).

Experiments:
- Please report standard errors or some other measure of variability for the results in Table 1. How many random seeds were considered?
- According to C1, you modified the implementation of the SOTA approach PCGrad to compute gradient w.r.t $z$, instead of all shared parameters as in the original paper. It is unclear how this modification affects PCGrad’s performance and if this choice maintains a fair comparison.
- Missing details of hyperparameters tuned for each method.


[1] Gradient Vaccine: Investigating and Improving Multi-task Optimization in Massively Multilingual Models, ICLR 2021.

[2] Understanding and Improving Information Transfer in Multi-Task Learning, ICLR 2020.




**Summary Of The Paper:**

This paper proposes a novel algorithm to tackle negative transfer in MTL optimization, by homogenizing the task gradients’ magnitude and direction. The method first homogenize gradients’ norm towards the norm of the task that converged the least. Next, it employs rotation matrices to align task gradients with the direction of the average gradient.


**Summary Of The Review:**

Overall it is a good paper with clear motivation and an intuitive approach. It tackles a major challenge in MTL optimization. Strong empirical results highlight the benefit of the approach compared to previous related methods.

---

> ### Author Response · Authors · 2021-11-12
> **Reply to Reviewer cExt**
>
> Dear Reviewer cExt,
>
> We are thankful for the great review and suggestions. Next, we provide clarifications to your specific comments:
>
> - Regarding scalability, please note that in §3.4 we present a time complexity analysis, showing that RotoGrad scales linearly in number of tasks and cubic in the matrix size $m$, $O(Km^3)$, as well as discuss ways to alleviate scaling issues. In the experiments, we specify the value of $m$ and, for the most complex datasets (NYUv2, CelebA), we also report training times in Tables 7 and 8 of the Appendix.
> - During early development, we did test other options for $v_n$. For example, as given by GradNorm, or by simply taking the average $\frac{1}{K} \sum_k g_k$. We opted for the average of unitary gradients $v = \frac{1}{K} \sum_k u_k$ as it is simple and works well in practice. Our intuition is that it provides a consistent direction that is not affected by gradient magnitudes, and thus RotoGrad can smoothly converge to it. If the reviewer considers these additional experiments to be relevant, we can run and add them to the Appendix.
> - We apologize for missing out these references, and thank the reviewer for the suggestions. We will add them to the next revision.
> - For the NYUv2 experiments, as they are computationally demanding, we consider a single seed. We will run more seeds and, while we may not make it to the end of the rebuttal, we will make sure the results are updated for the revised version.
> - Regarding the details of PCGrad in the feature space $z$, we rely on the same implementation as previous works (see, for example, [1] and [2]). We do so for fair comparison across methods, as all baselines modify the gradients wrt the shared feature $z$. Note that during backpropagation, the actual sum of gradients is performed in $z$, and thus PCGrad can be applied there to reduce the conflict at that level. We will clarify the details of this implementation in the Appendix.
> - We will add the hyperparameters details for all methods to §C1 on the following revision. We apologize for having missed these details in the current version.
>
> [1] Just Pick a Sign: Optimizing Deep Multitask Models with Gradient Sign Dropout - NeurIPS 2020
>
> [2] Towards Impartial Multi-task Learning - ICLR 2021

---

> > ### Comment · Reviewer_cExt · 2021-11-16
> > **Reply to authors**
> >
> > I thank the authors for the clarifications and the additional information.
> >
> > Regarding the NYUv2 experiment: results with a single seed are practically meaningless and can probably produce any ordering among the different modeling. I highly encourage the authors to update the results with additional random seeds and share them with the reviewers by the end of the rebuttal (even partial results due to time limitations).

---

> > > ### Author Response · Authors · 2021-11-16
> > > **Reply to Reviewer cExt. Re: NYUv2 experiments**
> > >
> > > Dear reviewer, we agree on the importance of running experiments on different seeds in order to avoid spurious conclusions. As promised, NYUv2 experiments on more random seeds are already running (we did not include the `RotoGrad+` rows showed in Table 8 due to time constraints).
> > >
> > > We are positive that we will manage to update Table 1 by the end of the rebuttal period. Moreover, we will try to implement as many of the other promised modifications to the manuscript as possible by the same date.

---

> > > > ### Comment · Reviewer_cExt · 2021-11-24
> > > > **Reply to authors**
> > > >
> > > > I thank the authors for adding the additional experimental results and updating the manuscript according to the reviewers' concerns. The revised manuscript addresses my main concerns, and I would like to maintain my initial acceptance score.

---

### Official Review · Reviewer_uDos · 2021-11-07

**Correctness:** 4
**Technical Novelty And Significance:** 4
**Empirical Novelty And Significance:** 3
**Recommendation:** 8
**Confidence:** 4

**Main Review:**

The proposed optimization scheme appears to be a unique technical approach among a larger set of recent work that attempt to deconflict gradients in multi-task learning.  Prior work has, e.g., focused on gradient projection operations [Yu et al., 2020], which, while similar in motivation, may lack the flexibility of the proposed approach's learnable rotations.

Experiments demonstrate proof-of-concept on synthetic optimization problems, as well as good results on real datasets (CIFAR10, CelebA, NYUv2).  Results on NYUv2 (semantic segmentation, depth estimation, normal estimation) look especially promising in comparison to both a baseline system and recently-published multi-task optimization procedures.

Overall, the proposed RotoGrad approach seems quite promising and could become a standard tool for multi-task neural network training.  It would be interesting to see results on larger-scale problems (both larger networks and larger datasets), as well as examine how RotoGrad might be applicable in transfer-learning and meta-learning scenarios.


**Summary Of The Paper:**

This paper introduces a training algorithm for neural networks that reduces conflict between multiple output tasks that utilize a shared portion of the network.  Considering a common shared representation from which multiple task-specific subnetworks branch, the approach aims to homogenize task-specific gradient magnitudes and directions at this branch point.  The proposed strategy is to introduce additional parameterized rotation matrices, each of which modifies the shared representation before it is passed to a corresponding task-specific branch.  The parameters of these rotation matrices are optimized to maximize gradient similarity between different tasks at the branch point; this optimization step is interlaced with standard updates of other network parameters to minimize total task loss.

**Summary Of The Review:**

This paper presents a parameterization scheme and optimization procedure for improved training of shared multi-task networks.  The technical approach is concise, well-motivated, and novel with respect to existing work.  Experiments across synthetic and real datasets show convincing results.

---

> ### Author Response · Authors · 2021-11-12
> **Reply to Reviewer uDos**
>
>
> Dear Reviewer uDos,
>
> Thank you for the thorough review and kind words. We would be pleased if RotoGrad can set the grounds for a new direction of research on negative transfer.
>
> While we agree that testing RotoGrad on bigger networks can be desirable, please note that the size of both datasets (NYUv2, CelebA, etc), and architectures (Resnet18, 6-layer CNNs, SegNet, etc) are on par with recent works on negative transfer in MTL (e.g. [1] and [2]).
> We would like to highlight that while the focus on this work is multitask learning, it would be indeed very interesting to apply RotoGrad to the frameworks mentioned by the reviewer. We defer these applications to future work.
>
> [1] Gradient Surgery for Multi-Task Learning - NeurIPS 2020
>
> [2] Multi-task learning as multi-objective optimization - NeurIPS 2018

---

### Author Response · Authors · 2021-11-17
**Preliminary NYUv2 experiments**

Dear reviewers and AC,

Here, we provide an update of Table 1 where now we show the median over 3 different random seeds, instead of the results of a single seed. The conclusions drawn in the manuscript can still be fully appreciated in this new table, with mostly subtle changes. The biggest difference regards the depth task, which highly fluctuated on the single-task baseline, and therefore the high values for $\Delta_D$.

We apologize for the formatting, we are restricted to markdown for this comment. We have highlighted the most interesting results to improve readability.

As a final note, let us remind the reviewers that we are waiting for two other seeds (a total of five) to finish before updating the manuscript. We will try to have them before the rebuttal deadline, as well as to implement as many of the other changes as possible before said deadline.

|      Method      | $\Delta_S$ | $\Delta_D$ | $\Delta_N$ | Seg. mIoU | Seg. PixAcc | Dep. Abs | Dep. Rel | Nor. Mean | Nor. Median | Nor. 11.25 | Nor. 22.5 |  Nor. 30  |
| :--------------: | :--------: | :--------: | :--------: | :-------: | :---------: | :------: | :------: | :-------: | :---------: | :--------: | :-------: | :-------: |
|      Single      |    0.00    |    0.00    |    0.00    |   37.67   |    63.46    |   0.71   |   0.28   |   25.09   |    19.18    |   30.01    |   57.33   |   69.30   |
|   Rotate Only    |  **3.93**  | **24.88**  | **-6.56**  | **39.71** |  **66.45**  | **0.52** | **0.21** |   26.12   |    20.93    |   26.85    |   53.76   |   66.50   |
|    Scale Only    |    2.07    |   17.86    |   -8.10    | **39.16** |    65.44    | **0.53** |   0.22   |   26.35   |    21.16    |   26.72    |   53.24   |   65.96   |
|     RotoGrad     |  **3.39**  | **24.04**  | **-6.11**  | **39.32** |  **66.07**  |   0.54   | **0.21** | **25.90** |  **20.80**  | **27.18**  | **54.04** | **66.75** |
| Vanilla + $R_k$  |    0.78    |   22.58    |  *-26.30*  |   38.05   |    64.39    |   0.54   |   0.22   |   30.02   |    26.16    |   19.95    |   43.44   |   56.87   |
| GradDrop + $R_k$ |  *-2.33*   |   15.95    |  *-25.38*  |   37.90   |    63.51    |   0.58   |   0.23   |   29.83   |    25.81    |   20.05    |   44.04   |   57.48   |
|  PCGrad + $R_k$  |  *-2.67*   |   20.47    |  *-26.31*  |   36.39   |    62.94    |   0.55   |   0.22   |   29.87   |    25.95    |   20.43    |   43.95   |   57.22   |
| MGDA-UB + $R_k$  |  *-31.23*  |  *-0.65*   |  **0.56**  |   21.60   |    51.60    |   0.77   |   0.29   | **24.80** |  **19.04**  | **30.29**  | **57.64** | **69.64** |
| GradNorm + $R_k$ |   -0.48    |   19.50    |   -10.45   |   37.22   |    63.61    |   0.57   |   0.22   |   26.66   |    21.67    |   26.02    |   52.16   |   64.95   |
|  IMTL-G + $R_k$  |    1.26    |   17.48    |   -8.27    |   38.38   |    64.66    |   0.55   |   0.22   |   26.38   |    21.35    |   26.56    |   52.84   |   65.69   |
|     Vanilla      |   -0.62    |   15.28    |  *-25.74*  |   37.11   |    63.98    |   0.58   |   0.22   |   29.93   |    25.89    |   19.85    |   43.92   |   57.39   |
|     GradDrop     |   -0.63    |   15.38    |  *-28.10*  |   37.11   |    62.88    |   0.59   |   0.23   |   30.47   |    26.63    |   19.01    |   42.61   |   56.06   |
|      PCGrad      |    1.50    |   19.97    |  *-24.63*  |   38.51   |    63.95    |   0.54   |   0.22   |   29.79   |    25.77    |   20.61    |   44.22   |   57.64   |
|     MGDA-UB      |  *-32.75*  |  *-8.22*   |  **1.50**  |   20.40   |    51.36    |   0.73   |   0.28   | **24.70** |  **18.92**  | **30.57**  | **57.95** | **69.99** |
|     GradNorm     |  **3.59**  |   21.72    |   -10.23   | **39.53** |    65.27    | **0.53** |   0.22   |   26.77   |    21.88    |   25.39    |   51.78   |   64.76   |
|      IMTL-G      |    2.68    |   22.48    |   -7.43    | **39.94** |    65.56    |   0.55   | **0.21** |   26.23   |    21.14    |   26.38    |   53.25   |   66.22   |

---

### Author Response · Authors · 2021-11-22
**Updated manuscript**

Dear reviewers and AC,

We would like to inform you that we have updated the manuscript, trying to incorporate as much feedback as possible. The most important changes are the following:
-  We have run the NYUv2 experiments of 5 different seeds for all methods (including those with `RotoGrad +` in the appendix). We have updated Table 1 and Table 8 accordingly. In Table 8 we included all training times (which were missing before due to inconsistencies with our cluster configuration).
- For all tables with several runs, we have run a paired one-sided t-test with confidence threshold of 0.05, and highlighted in the tables (main text and appendix) those values that are significantly better than multitask learning with vanilla optimization.
- We have expanded related work to include those references provided by the reviewers, while keeping the limit of 9 pages for the manuscript.
- As suggested, the abstract has been slightly modified to improve the grammar.
- Hyperparameters for all methods are explained in section C.1 of the appendix (`Methods hyperparameters` paragraphs).
- The implementation of PCGrad at the feature level has been better explained in section C.1 (`Baselines` paragraph).

We are working on further improving the manuscript, including those experiments involving the CelebA dataset, which we hope we can update before the rebuttal deadline.

---

### Author Response · Authors · 2021-11-23
**New manuscript**

Dear reviewers and AC,
We are pleased to announce that we have now uploaded a final update to the manuscript. In this new version, we have updated the results on CelebA, where we now test all methods (but MGDA-UB, for computational reasons) on five different random seeds, instead of on a single one. RotoGrad keeps performing really well in both architectures, and specially in the convolutional network.

As a result of the newly introduced experiments, we have also been able to provide training times for all NYUv2/CelebA experiments in the appendix, by taking the median over all the runs.

Thanks,
RotoGrad's authors

---

### Decision · Program_Chairs · 2022-01-20

**Decision:**

Accept (Spotlight)

**Comment:**

The paper addresses the problem of inconsistent gradients in multi-task learning, proposing ways to handle both their magnitude nd direction. Gradient directions are aligned by introducing a rotation layer between the shared backbone and task-specific branches.
Reviewers appreciated the technical approach, higlighting the novelty of the rotation layers in this context. The empirical evaluations are systematic fair and insightful, and the presentation is polished. Reviewers unanimously supported accepting the paper.